# SimUSER: When Language Models Pretend to Be Believable Users in Recommender Systems

## Abstract

Recommender systems play a central role in numerous real-life applications, yet evaluating their performance remains a significant challenge due to the gap between offline metrics and online behaviors. We introduce SimUSER, an agent framework that serves as believable and cost-effective human proxies for the evaluation of recommender systems. Leveraging the inductive bias of foundation models, SimUSER emulates synthetic users by first identifying self-consistent personas from historical data, enriching user profiles with unique backgrounds and personalities. Then, central to this evaluation are users equipped with persona, memory, perception, and brain modules, engaging in interactions with the recommender system. Specifically, the memory module consists of an episodic memory to log interactions and preferences, and a knowledge-graph memory that captures relationships between users and items. The perception module enables visual-driven reasoning, while the brain module translates retrieved information into actionable plans. We demonstrate through ablation studies that the components of our agent architecture contribute to the believability of user behavior. Across a set of recommendation domains, SimUSER exhibits closer alignment with genuine humans than prior state-of-the-art, both at micro and macro levels. Additionally, we conduct insightful experiments to explore the effects of thumbnails on click rates, the exposure effect, and the impact of reviews on user engagement. The source code is released at https://github.com/SimUSER-paper/SimUSER.

## 1 Introduction

Recommender systems (RS) have become an indispensable component of our day-to-day lives, offering personalized user experience and improving satisfaction Li et al. (2024b). By examining granular preferences, historical data, and contextual factors, RS can deliver recommendations that are tailored to meet individual tastes. Despite their widespread adoption, a key challenge hindering the advancement of the field is evaluation Yoon et al. (2024). The difficulty arises from the discrepancy between offline metrics (non-interactive), which are typically used during development, and real-life user behaviors, which these systems encounter post-deployment Zhang et al. (2019). This results in models that perform well in controlled environments but fail to meet expectations in practical use cases. Such a limitation is further exacerbated by the inherent shortcomings of offline evaluation, notably the inability to measure business values such as user engagement and satisfaction Jannach & Jugovac (2019). On the other hand, deploying and testing RS in real-world settings is both costly and labor-intensive, underscoring the imperative need for reliable and affordable (interactive) evaluation methods.

Recent breakthroughs in Large Language Models (LLMs) have shown promise in human behavior modeling by enabling the creation of autonomous agents. Generative agents have demonstrated capabilities in lifelong learning through automatic skill discovery Wang et al. (2023a) or setting consistent goals during exploration Du et al. (2023). LLMs have also been applied to various social simulations, such as hospital Li et al. (2024a) and city Park et al. (2023). In a different spirit, $S^3$ Gao et al. (2023) simulates the dynamic evolution of opinions in a social network. In the realm of recommendation systems, RecMind Wang et al. (2023e) explores the concept of autonomous recommender agents equipped with self-inspiring planning and external tool utilization. Recently, InteRecAgent Huang et al. (2023) has extended this idea by proposing memory components, dynamic demonstration-augmented task planning, and reflection. Nevertheless, their primary focus

remains on identifying candidate items that align with user preferences. In addition, these agents are overly simplified representations of human users, limited to predicting item ratings based on viewing history. Recently, RecAgent Wang et al. (2023c) has attempted to introduce more diverse user behaviors, taking into account external social relationships. Another work, Agent4Rec Hou et al. (2024), delves into generating faithful user-RS interactions via agent-based simulations, where agents are equipped with a memory module. However, a common characteristic of existing studies is their *insulated nature* — they primarily rely on knowledge embedded within the model's weights, neglecting the potential benefits of integrating external knowledge and user-item relationships. Furthermore, prior approaches often disregard user personas (e.g., personality traits) and fail to incorporate visual signals, despite images significantly shaping user experience and emotion.

To enable synthetic users, this paper describes an agent architecture built upon large language models. Our methodology consists of two phases: (1) self-consistent persona matching and (2) recommender system evaluation. In Phase 1, we leverage the semantic awareness of LLMs to extract and identify consistent personas from historical data, encompassing unique backgrounds, personalities, and characteristics. In Phase 2, we impersonate these identified personas to simulate believable human interactions. This involves a retrieval-augmented framework where the agent interacts with the recommender system based on its persona, memory, perception, and brain modules. The memory module comprises an episodic memory and a knowledge-graph memory. The former is the episodic memory that records, in natural language, a comprehensive list of the agent's experiences. The latter captures relationships between users and items, accounting for the influence of other users and prior beliefs about items. Unlike existing studies that solely rely on text, our perception module incorporates visual cues into the agent's reasoning process. Finally, the brain module is responsible for translating retrieved information and the current simulator observation into action plans such as `watch`, `click`, or `exit`. Following action selection, the user engages in self-reflection to synthesize memories into higher-level inferences and draw conclusions.

To our knowledge, this study is one of the pioneering works in developing a general LLM-based agent framework for systematic evaluation of recommender systems, integrating persona extraction, visual-driven reasoning, and domain-specific prior knowledge to simulate realistic user interactions.

## 2 RELATED WORK

**LLMs as Human Proxies.** The use of LLMs to replace costly human trials is becoming increasingly popular. For instance, Owoicho et al. (2023) and Wang et al. (2024d) examine LLMs as user simulators for conversational search, while Hämäläinen et al. (2023) seeks to generate synthetic user experience data in human-computer interaction (HCI). Significant progress have also been achieved in the area of collaborative cooperation. For instance, CAMEL Li et al. (2023a) describes a framework for agent cooperation that features a commander for planning and executors for task implementation. Qian et al. Qian et al. (2023) introduces a virtual software company where agents, each assigned roles such as engineer, work together to complete software development projects. Besides, Aher et al. (2023) explores LLMs' ability to replicate human behavior for social science experiments. Other researchers have created simulation environments where LLM agents interact with one another, producing realistic daily activities Park et al. (2023); Gao et al. (2023) or generating mobility trajectories Wang et al. (2024b). Nonetheless, their potential for assessing recommender systems and addressing inherent issues in recommendation remains largely unexplored.

**Simulating Users in Recommendation.** Conversational RS initially tackled the recommendation problem using bandit models, emphasizing the quick update of traditional systems through item selection and binary feedback from synthetic users Christakopoulou et al. (2016). Taking this further, Zhao et al. (2023) created a simulation platform where users not only chat about recommendations but also navigate websites, search for items, and share opinions on social media. Recent techniques have added more natural language flexibility, but user responses are usually limited to binary or multiple-choice formats Lei et al. (2020). Evaluations typically involve predefined target items for each user, with success measured by the number of turns needed to identify these targets Guo et al. (2018); Sun & Zhang (2018). Another technique employs agenda-based simulations, utilizing state diagrams to guide actions, and deeming a recommendation successful when the conversation reaches a "complete" state Zhang & Balog (2020); Zhang et al. (2022). In spite of this, these simulations often rely on fixed rules and scripted dialogues, lacking the variability seen in human interactions.

**LLMs in Recommender Systems.** To address the above-mentioned limitations, generative simulators using Large Language Models (LLMs) have been developed, offering more realistic and nuanced conversational abilities Zhang et al. (2024b); Zhao et al. (2023). A few studies have also explored the application of LLMs as recommender systems Hou et al. (2024); Li et al. (2023b); Kang et al. (2023); Fan et al. (2023); Yang et al. (2023a). These investigations explore LLMs as recommendation engines, rather than as entities that perceive recommendations, thus providing a perspective complementary to our research Wang et al. (2024c); Zhang et al. (2024a). Some authors Wang et al. (2023d) utilize ChatGPT to prompt users towards specific items, where users would provide 'hints' about these items. LLM4Rec Wang et al. (2024a) alleviates the beam search decoding by using a straight item projection head for ranking scores generation, providing speedup when predicting the next item. Rec-GPT4V investigates the use of vision-language models for multimodal recommendation Liu et al. (2024). In order to bridge the gap between reasoning capabilities and external knowledge, KAR Xi et al. (2023) incorporates open-world knowledge, which includes factual knowledge on items and user preferences. An emerging line of work seeks to simulate user behaviors in the context of RS. For instance, RecAgent Wang et al. (2023b) presents a new simulation framework aimed at enhancing the evaluation of recommendation algorithms through agent-based simulation. In a similar fashion RecMind Wang et al. (2023e) proposes self-inspiring agents for recommendation. However, their simulated users are limited to basic actions like rating items, lacking the ability to engage in more complex interactions, such as navigating the interface. Notably, a recent approach Yoon et al. (2024) examines the effectiveness of LLMs as generative users, specifically for conversational recommendation scenarios. A closely related work to ours is Agent4Rec Zhang et al. (2023) that delves into the generative capabilities of LLMs for modeling user interactions. SimUSER differs significantly from these studies in several key aspects. We simulate realistic humans using detailed personas that are inferred from their interaction history. We also incorporate a perception module to integrate visual information in decision-making, mapping thumbnail items to token embeddings of the language model. Furthermore, the present study explores the potential for graph-based retrieval and features a richer spectrum of actions such as "going to previous page" or "clicking", to better replicate influential factors and user experience.

## 3 METHODOLOGY

**Sim**ulated **USER**s provides a framework for systematically assessing recommender systems by engaging in interactions and providing feedback. As illustrated in Figure 1, we present a two-phase method. Phase 1 of SimUSER matches historical data with a set of personas to enable nuanced and realistic interactions with the recommender system. Phase 2 utilizes the identified personas, historical data, and novel reasoning mechanisms to generate synthetic users with human-like behavior. At the center of our architecture is the memory module. From the memory module, interaction records and prior assumptions are retrieved to condition the agent's actions and react appropriately to the simulation. Additionally, the perception module enables agents to transform visual cues into a natural language form when interacting with the recommender system.

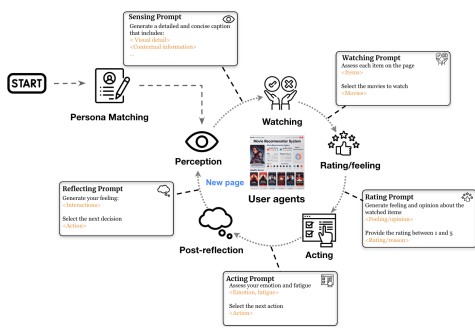

Figure 1: Illustration of SimUSER framework.

**Problem Formulation.** Given a user $u \in \mathcal{U}$ and an item $i \in \mathcal{I}$, the aggregated rating of the item is denoted by $R_i = \frac{1}{\sum_{u \in \mathcal{U}} y_{ui}} \sum_{u \in \mathcal{U}} y_{ui} \cdot r_{ui}$ where $y_{ui} = 0$ indicates that the user $u$ has not rated the item $i$ and inversely $y_{ui} = 1$ indicates that the user has rated the item with $r_{ui} \in \{1, 2, 3, 4, 5\}$. We also introduce $g_i \in G$ as the genre/category of the item. In this study, we seek to discover $y_{ui}$ and $r_{ui}$ for an unseen recommended item $i$.

**Generative Large Language Models.** LLMs are trained to predict the most probable next token $t_k$ given the sequence of previous tokens $t_1 \dots t_{k-1}$ by maximizing the likelihood function $p_{LLM}(t_k | t_1, \dots, t_{k-1})$. In this work, we use pre-trained LLMs without further finetuning them.

Depending on the task, we generate one or more tokens given a task-specific context $c^{(p)}$ that describes the task to the language model and prompts it for an answer. Thus, we obtain generated tokens $\mathbf{t}$ by sampling from:

$$p_{LLM}(t|c^{(p)}) = \prod_{k=1}^{K} p_{LLM}(t_k|c_1^{(p)}, \ldots, c_n^{(p)}, t_1, \ldots, t_{k-1}) \quad (1)$$

## 3.1 Persona Matching via Self-Consistency Check

This phase involves assessing the most plausible *persona* based on historical data. A persona $p$ encompasses a set of features that characterize the user: **age**, **personality**, and **occupation**. Personality traits are defined by the Big Five personality facets: *Openness*, *Conscientiousness*, *Extraversion*, *Agreeableness*, and *Neuroticism*, each measured on a scale from 1 to 3. We postulate that these factors are critical for modeling personalized preferences. Given the difficulty of obtaining such granular features in real-world settings, our methodology seeks to systematically infer personas from the user's interaction history.

**Persona Extraction.** Assuming a user $u$ and a set of interactions $\{(i_0, r_{ui_0}), (i_1, r_{ui_1}), \ldots, (i_n, r_{ui_n})\}$, we first query the LLM to produce a short summary $s_u$ of the user's preferences. To do so, we randomly select 50 items from the user's viewing history. Items rated 4 or above are categorized as *liked*, while those rated below 3 are deemed *disliked*. This summary describes its unique tastes and rating patterns.

Following this, both the summary $s_u$ and historical data are combined in the prompt, instructing the LLM agent to generate a persona that matches the interaction history for this user. To enhance the diversity in the generated personas, the LLM is provided a list of possible ages, personalities, and occupations. For each user, a set of $m$ ($m = 5$) candidate personas is generated, denoted as $\mathcal{P}$.

**Self-Consistent Persona Evaluation.** We then assess the consistency of the candidate personas $\mathcal{P}$ to identify the most plausible one. Specifically, a self-consistency scoring mechanism measures the alignment of candidate personas with historical data. We define a scoring function $s(p, u)$ for each candidate persona $p \in \mathcal{P}$, where $p$ is evaluated against two distinct sets of user-item interactions. For the targeted user $u$, we sample $j$ subsets of $\varrho$ interactions from its history. These are compared with $\varrho$ sampled interactions from other users $\bar{u}$, denoted as $I_{\bar{u}}$:

$$s(p, u) = \sum_{\iota \sim I_u} \hat{r}(\iota, p) - \sum_{\bar{\iota} \in I_{\bar{u}}} \hat{r}(\bar{\iota}, p) \quad (2)$$

where $\hat{r}(\iota, p)$ and $\hat{r}(\bar{\iota}, p)$ are obtained by querying the LLM to rate the two interaction subsets $\iota$ and $\bar{\iota}$. Ideally, the LLM agent should assign a higher $\hat{r}(\iota, p)$ for interactions from the targeted user and a lower $\hat{r}(\bar{\iota}, p)$ for samples from other users. This approach ensures self-consistency in the persona: the persona derived from the rating history should be consistent with the data during the evaluation. The candidate persona $p$ with the highest score $s$ is assigned to the user. We provide the prompts for Phase 1 in Appendix B.

## 3.2 Engaging in Interactions with Recommender Systems

In Phase 2, given a user $u$ and its discovered persona $p$, we present a novel cognitive architecture built upon LLMs to simulate faithful human proxies. This model accounts for various factors, including the user's persona, memory of past interactions, habits, and unique tastes/preferences. It comprises four major modules: **persona**, **perception**, **memory**, and **action**.

### 3.2.1 Persona Module

In the recommendation domain, the user's persona plays a central role in aligning the agent's behavior with genuine human actions Li et al. (2024b). To lay a reliable foundation for the generative agent's subsequent interactions and evaluations, benchmark datasets (e.g., MovieLens-1M Harper & Konstan (2015), Steam Kang & McAuley (2018), AmazonBook McAuley et al. (2015)) are used for initialization of the persona module. In detail, each agent's profile includes the matched persona $p$ along with attributes extracted from its historical data: $p \cup \{$**pickiness**, **habits**, **unique tastes**$\}$.

Since LLMs are biased towards positive sentiment, unless prompted to behave as picky users Yoon et al. (2024), each agent is assigned a *pickiness* level sampled in {*not picky*, *moderately picky*, *extremely picky*} based on the user's average rating.

Habits, as defined in Zhang et al. (2023), account for user tendencies in interactions with recommender systems: engagement, conformity, and variety.

- Engagement quantifies the frequency and breadth of a user's interactions with recommended items, distinguishing between users who extensively watch and rate many of items and those who confine themselves to a minimal set. The engagement trait for user $u$ can be mathematically expressed as: $T^u_{act} = \sum_{i \in \mathcal{I}} y_{ui}$.

- Conformity measures how closely a user's ratings align with average item ratings, drawing a distinction between users with unique perspectives and those whose opinions closely mirror popular sentiments. For user $u$, the conformity trait is defined as: $T^u_{conf} = \frac{1}{\sum_{i \in \mathcal{I}} y_{ui}} \sum_{i \in \mathcal{I}} y_{ui} \cdot |r_{ui} - R_i|^2$.

- Variety reflects the user's proclivity toward a diverse range of item genres or their inclination toward specific genres. The variety trait for user $u$ is formulated as: $T^u_{div} = |U_{i \in \{y_{ui}=1\}} g_i|$. To encode users' unique tastes in natural language, we utilize the summary $s_u$ obtained in Phase 1, which describes long-term preferences.

Unique tastes of the user, derived from their viewing history, are encapsulated in the summary $s_u$ generated in Phase 1. This summary offers a detailed overview of their preferences, including both favored and disliked genres, and watching habits.

### 3.2.2 PERCEPTION MODULE

A primary factor in decision-making is visual stimuli due to their significant influence on curiosity and emotion Liu et al. (2024). For instance, when scrolling through a movie recommendation platform, human decisions are heavily driven by the thumbnails of items, which can trigger emotional responses and provide quick visual summaries of the content Koh & Cui (2022). To graft these visual elements in an cost-efficient manner, we propose augmenting action prompts (see Sec 3.3) with image-derived captions. The caption $i_{caption}$ of an item $i$ is generated by querying GPT-4o to extract insights that specifically capture emotional tones, visual details, and unique selling points from the item's thumbnail. The captions $i_{caption} = LLM(P_{caption}, i)$ are generated only once for each item before the interactions begin. For the sake of simplicity, all agents share the same captions. In future work with richer user interface, we anticipate using the raw images as input for a multimodal LLM or conditioning the captioning prompt $P_{caption}$ with the user's persona.

### 3.2.3 MEMORY MODULE

It is of vital importance for an agent to maintain a memory of the knowledge and experience it has of the world and others. We propose to use an episodic memory storing the interactions with the RS and a knowledge-graph memory that leverages graph-structured data to capture relationships, enabling the model to access knowledge beyond what is inherently present in the LLM's parameters.

▢ **Episodic Memory** retains a comprehensive record of the user's interactions with the recommendation system, including ratings, liked and disliked items, as well as the underlying reasons. For instance, an entry might be "*You rated a movie called 'Star Wars: Episode VI' 4 out of 5*". The memory is initially populated with the user's viewing and rating history. Each time SimUSER executes a new action or rate an item, the corresponding interaction is added to the episodic memory. At inference time, the agent conditions its decisions on information retrieved from this memory.

For humans, recall is the psychological process of accessing memories from the past Atkinson & Shiffrin (1968). In our system architecture, it involves retrieving "documents" from the episodic memory. We implement a self-ask strategy by prompting the LLM to raise follow-up questions regarding the query. These questions, along with the initial query, then serve as queries for vector similarity search. Given a query $q$, the top-$k_1$ documents $d$ that have the highest similarity scores $s(q, d)$ when compared with the query are retrieved in this step:

$$s(q, d) = \cos(\mathbf{E}(q), \mathbf{E}(d)) = \frac{\mathbf{e}_q \cdot \mathbf{e}_d}{||\mathbf{e}_q|| ||\mathbf{e}_d||} \tag{3}$$

where $\mathbf{E}$ is an embedding function.

❋ **Knowledge-Graph Memory** User behaviors in real-life recommender systems are shaped by both internal and external factors Zhao et al. (2014). Examples of internal factors include user personality, habits, and age, which are captured in the persona module. For external factors, we consider the influence of other users and prior beliefs about items. To simulate external influences, SimUSER leverages a KG memory that is used to retrieve items sharing similar relationships within the graph and their intrinsic characteristics. The retrieval process emulates the way we form beliefs, influenced by recommendations from friends, family, and prior knowledge, which is crucial for reducing hallucinations and contextualizing user preferences.

**Memory Initialization** The knowledge graph memory is initially populated using real-world datasets such as MovieLens, AmazonBook, and Steam. It is structured as a **K**nowledge **G**raph (KG) $\mathcal{G}$ where nodes denote entities, and edges correspond to relations between entities. An edge in the KG represents a fact stored in the form of *(subject, predicate, object)*. Formally, the knowledge graph $\mathcal{G} = \{(h, r, t)|h, t \in \mathcal{V}, r \in \mathcal{E}\}$, in which each triple *(h,r,t)* indicates that a relation $r$ exists from head entity $h$ to tail entity $t$. $\mathcal{V}$ is a set of entities and $\mathcal{E}$ represents relationships between them. For instance, nodes $\mathcal{V}$ may represent entities such as *users* and *items*, while edges $\mathcal{E}$ may depict the relationships between these entities such as such as *liked*.

**Memory Growth** The memory grows with each interaction, capturing the evolving nature of user preferences and behaviors. Formally, given an interaction $i_t$ at time $t$, such as liking, disliking, or rating an item, the memory update can be represented as:

$$\mathcal{G}_{t+1} = \mathcal{G}_t \cup \{(v_i, e_{ij}, v_j)|(v_i, e_{ij}, v_j) \in \mathcal{V} \times \mathcal{E} \times \mathcal{V}\} \tag{4}$$

where $v_i$ and $v_j$ are entities (nodes) involved in the interaction $i_t$ (e.g., user and item) and $e_{ij}$ is the relationship (edge) created or updated by the interaction. For simplicity, this scheme assumes that agents only perceive their own interactions.

**Graph-Aware Dynamic Item Retrieval** For a user $u$, the retrieval function takes a query item $x$ as input and returns a set of similar items along with their associated metadata (e.g., *ratings*), leveraging the KG structure and semantic similarity. A variety of algorithms may be used to measure the similarity between two items. In our pipeline, we extend Pathsim Sun et al. (2011) on account of its flexibility to capture both user-item and item-item relationships.

A relationship path $p_{x \rightsquigarrow y}$ represents a composite relationship between entities $x$ and $y$ in the form of $x \xrightarrow{\mathcal{E}_1} z \xrightarrow{\mathcal{E}_2} \dots \xrightarrow{\mathcal{E}_l} y$, where $\mathcal{E}_1$ denotes the edge between entity $x$ and $z$. For example, in the MovieLens network, the co-actor relation can be described using the length-2 relationship path $x \xrightarrow{acts-in} z \xrightarrow{actor} y$. In order to retrieve relevant items based on the query $x$, SimUSER estimates the item-item similarity as follows:

$$s_{x,y} = \frac{2 \times \left|\{p_{x \rightsquigarrow y} : p_{x \rightsquigarrow y} \in \mathcal{P}\}\right|}{\left|p_{x \rightsquigarrow x} : p_{x \rightsquigarrow x} \in \mathcal{P}\right| + \left|p_{y \rightsquigarrow y} : p_{y \rightsquigarrow y} \in \mathcal{P}\right|} \tag{5}$$

where $\mathcal{P}$ denotes the set of paths between the query item $x$ and candidate item $y$, and $p_{x \rightsquigarrow y}$ is a path instance. The score $s_{x,y}$ is determined by two factors: (1) the connectivity level, which is the count of paths that connect $x$ and $y$ through $\mathcal{P}$; and (2) the balance of visibility, defined by the number of times these paths are traversed between the two entities. In addition to item-item similarity $s_{x,y}$, we compute user-item similarity $s_{u,y}$ for the target user $u$ and the candidate item $y$, using the same path-based approach, which is further summed up to $s_{x,y} = \alpha \cdot s_{x,y} + (1 - \alpha) \cdot s_{u,y}$. Thus, the retrieval output is influenced by the communities surrounding both the current user and the target item, along with past interactions of $u$.

To account for the semantic similarity between items, the weighted cosine similarity $\frac{\mathbf{e}_x \cdot \mathbf{e}_y}{\|\mathbf{e}_x\|\|\mathbf{e}_y\|}$ is summed up with $s_{(x,y)}$, where $\mathbf{e}_x$ is the node embedding generated with OpenAI's model *text-embedding-3-small*. We found the semantic score helpful in capturing hidden variables that are not represented in the KG. Finally, the top-$k_2$ item candidates and their attributes are returned and serve to condition the brain module. Note that the `rating` of a retrieved item is substituted with the user's own rating if available; otherwise, it defaults to the average rating from other users stored in $\mathcal{G}$.

## 3.3 Brain Module

We endow each agent with a decision-making module that derives the agents' subsequent actions based on the retrieved information, considering their persona, tastes, emotions, and fatigue. It involves 1) **deciding** on items of interest, 2) **rating** them and **providing** feelings, 3) **selecting** the next action, and 4) **engaging** in post-interaction reflection. To replicate human-like sequential reasoning, we employ Chain-of-Thought prompting, repeatedly performing the four steps.

**Watching Items:** In our simulation, agents browse items page by page. According to their preferences, persona, and watching history, they assess each item on the page and may decide to "watch" certain items — agents select items that interest them. To address the inherent bias towards positive sentiment often observed in LLMs Yoon et al. (2024), SimUSER incorporates a pickiness modifier into the prompt: `You are {pickiness} about {item_type}`. When available, we enrich item descriptions with their thumbnail captions to enable multimodal reasoning. Finally, for each item $i$, the top-$k_2$ similar items and their ratings are retrieved from the knowledge graph memory, helping to mitigate LLM hallucinations — reducing the impact of any singular bias towards or against an item, while enriching context information.

**Providing Feelings and Rating Items** This action is triggered once the user has found the items of interest. Intuitively, a real user may produce much feelings after watching an item, which will be stored in their memory and influence their future cognition and behaviors. Along with the item rating $\in \{1, 2, 3, 4, 5\}$, we query the user's feelings about the watched items and leverage such information to update the memory module. That is, newly liked and disliked items are fed back into the memory module to influence the agent's future behavior. In order to enhance consistency, users are asked to provide a reason for their feedback.

**Emotion-driven Action Selection** Once the recommender system returns the search or recommendation results, including $n$ items each time, the agent discerns the current recommendation page and retrieves its interaction history. Drawing upon these insights combined with its persona, the agent decides whether to `[EXIT]` the system, go to `[NEXT]` page, return to a `[PREVIOUS]` page, or `[CLICK]` on an item to access more details. To do so, the agent sequentially: 1) estimates its satisfaction level with preceding recommendations, 2) generates its current fatigue level Zhang et al. (2023), 3) infers its current emotion, such as `EXCITED`, and 4) selects the most suitable action. Satisfaction level, fatigue, and emotion are dynamic elements that the agent employs to adapt its actionable plan with the recommender system. If the agent decides to click on an item, the item is displayed with an extended description that replaces the short $\{item\_description\}$ as outlined in Appendix B, which is then used to determine whether it wishes to engage further with the item. Finally, if `[EXIT]` is selected, a satisfaction interview is conducted to gather granular opinions and ratings on the presented recommendations.

**Post-interaction Reflection** LLM-Empowered agents, when equipped with only raw episodic memory, struggle to generalize or make inferences Park et al. (2023). Post-interaction reflection is a mechanism designed to let agent learn from interactions and improve future alignment with their persona. After the agent engages in assessing and rating the items on the page, we collect interaction data, including the items clicked and watched, along with the user's explicit feedback (e.g., ratings, feelings). The first step in reflection is for the agent to determine what to reflect on, then we prompt the language model to extract insights and cite the particular records that served as evidence for the insights. The post-interaction reflections are fed back into the episodic memory.

## 4 Experiments

### 4.1 Experimental Settings

**Datasets.** We investigate three real-world datasets: MovieLens-1M Harper & Konstan (2015), Steam Kang & McAuley (2018), and AmazonBook McAuley et al. (2015). They are employed for the initialization of each agent — persona and memory modules, as well as self-consistent persona matching. In order to address privacy concerns, the name and gender are omitted. Moreover, for the sake of generality, we do not utilize user-specific information available in these datasets, relying instead on the personas identified in Phase 1 of SimUSER.

**Settings.** All agents are powered by the GPT-4o-mini version of ChatGPT, except when specified

Table 1: User preference alignment across MovieLens, AmazonBook, and Steam datasets.

| Method(1:m) | MovieLens | | | | AmazonBook | | | | Steam | | | |
|---|---|---|---|---|---|---|---|---|---|---|---|---|
| | Accuracy | Precision | Recall | F1 Score | Accuracy | Precision | Recall | F1 Score | Accuracy | Precision | Recall | F1 Score |
| RecAgent (1:1) | 0.5807 | 0.6391 | 0.6035 | 0.6205 | 0.6035 | 0.6539 | 0.6636 | 0.6587 | 0.6267 | 0.6514 | 0.6490 | 0.6499 |
| RecAgent (1:2) | 0.5498 | 0.7475 | 0.5269 | 0.6178 | 0.6372 | 0.6520 | 0.5511 | 0.5970 | 0.6240 | 0.6788 | 0.5868 | 0.6290 |
| RecAgent (1:3) | 0.5077 | 0.7396 | 0.3987 | 0.5181 | 0.6144 | 0.6676 | 0.4001 | 0.5003 | 0.5873 | 0.6674 | 0.3488 | 0.4576 |
| RecAgent (1:9) | 0.4800 | 0.7491 | 0.2168 | 0.3362 | 0.6222 | 0.6641 | 0.1652 | 0.2647 | 0.5995 | 0.6732 | 0.1744 | 0.2772 |
| Agent4Rec (1:1) | 0.6912 | 0.7460 | 0.6914 | 0.6982 | 0.7190 | 0.7276 | 0.7335 | 0.7002 | 0.6892 | 0.7059 | 0.7031 | 0.6786 |
| Agent4Rec (1:2) | 0.6466 | 0.7602 | 0.5058 | 0.5874 | 0.6842 | 0.6888 | 0.5763 | 0.5850 | 0.6755 | 0.7316 | 0.5371 | 0.5950 |
| Agent4Rec (1:3) | 0.6675 | 0.7623 | 0.4562 | 0.5433 | 0.6707 | 0.6909 | 0.4423 | 0.5098 | 0.6505 | 0.7381 | 0.4446 | 0.5194 |
| Agent4Rec (1:9) | 0.6175 | 0.7753 | 0.2139 | 0.3232 | 0.6617 | 0.6939 | 0.2369 | 0.3183 | 0.6021 | 0.7213 | 0.1901 | 0.2822 |
| SimUSER (1:1) | **0.7834** | 0.7942 | **0.7455** | **0.7691** | **0.8012** | 0.7865 | **0.7538** | **0.7698** | **0.7809** | 0.7812 | **0.7651** | **0.7731** |
| SimUSER (1:2) | 0.7533 | 0.7811 | 0.5666 | 0.6566 | 0.7390 | **0.7887** | 0.6659 | 0.7216 | 0.7512 | **0.7956** | 0.6063 | 0.6883 |
| SimUSER (1:3) | 0.7412 | 0.7989 | 0.5124 | 0.6263 | 0.6512 | 0.7432 | 0.5542 | 0.6341 | 0.7299 | **0.7959** | 0.5248 | 0.6351 |
| SimUSER (1:9) | 0.6776 | **0.8096** | 0.3420 | 0.4794 | 0.6388 | 0.7460 | 0.3113 | 0.4384 | 0.7005 | 0.7720 | 0.2651 | 0.3934 |

differently, with the number of agents set to 1,000. The prompts and implementation details can be found in the Appendix B. Our analysis consists of two perspectives: micro-level and macro-level.

**Baselines** We compare the performance of SimUSER against RecAgent Wang et al. (2023c) and Agent4Rec Zhang et al. (2023), which represent the closest comparable methods for systematic recommender system assessment. When possible, we report the results of RecMind Wang et al. (2023e), an agent-based RS. Some experiments involve two versions of SimUSER: SimUSER(zero) and SimUSER(sim), where SimUSER(sim) agents first interact with the recommender platform — grounding interactions and filling their memories, before answering the tasks.

## 4.2 BELIEVABLY OF SYNTHETIC USERS

A central question is: How accurately do synthetic users replicate the behaviors of real humans? In order to appropriately respond to recommendations, synthetic users must possess a clear understanding of their own preferences. Thereby, we query the agents to classify items based on whether their human counterparts have interacted with them or not. Specifically, each of the 1,000 agents is randomly assigned 20 items. The proportion of items that the user has interacted with but was not used for persona and memory modules initialization to those that the user has not interacted with (represented as $y_{ui} = 0$) is set as $1 : m$ with $m \in \{1, 2, 3, 9\}$. We treat the problem as a binary classification task, taking values between 0 and 1. The results are presented in Table 1, from which we can see that SimUSER agents accurately identify items that are aligned with their own tastes. Across varying levels of distractors, SimUSER exhibits significantly superior performance compared to RecAgent and Agent4Rec, as indicated by paired t-tests at 95% confidence level (p < 0.002). Our method achieves an accuracy ranging from 78% to 80%, and recall between 26% and 76%, whereas Agent4Rec's accuracy spans from 60% to 71%, and recall varies from 19% to 73%. We attribute SimUSER's improvements to the use of a persona module augmented with granular information (e.g., age, occupation) obtained via persona matching. Notably, Agent4Rec displays a tendency to favor newly encountered items positively, resulting in a higher rate of item approval compared to their human counterparts. SimUSER overcomes this drawback by leveraging the KG memory to retrieve similar items, reducing bias towards unfamiliar items.

## 4.3 RATING ITEMS

A key task when interacting with a recommender system is rating items — predicting the rating that a user would give to a particular item. We compare the results of several LLM-based baselines, RecMind-SI (few-shot) Wang et al. (2023e), RecAgent, and Agent4Rec, along with traditional recommendation baselines: Matrix Factorization (MF) Koren et al. (2009) and Attentional Factorization Machines (AFM) Xiao et al. (2017). The results of rating prediction are summarized in Table 2. Across all tasks, we observe that the present study considerably outperforms the other LLM-powered agents: RecAgent, Agent4Rec, and RecMind-SI. Such improvement mainly stems from the advantage that SimUSER has access to the knowledge-graph memory that encapsulates priors about items and their relationships with user interactions. In contrast, Agent4Rec usually has a much higher RMSE, which can be attributed to its hallucination when rating items not embedded within its LLM's weights, particularly niche items. As expected leveraging persona matching, as done in SimUSER(persona), reduces the prediction errors compared to SimUSER(no-persona). Another interesting trend is that incorporating a few steps of simulation always decreases the MAE of

Table 2: Performance comparison in rating prediction on MovieLens, AmazonBook, and Steam datasets. The best results of each model are marked in **bold** and the second-best results are marked with underline. The improvement of SimUSER over all baselines on overall performance is statistically significant (measured by student's t-test at $p < 0.05$).

| Methods | MovieLens | | AmazonBook | | Steam | |
|---|---|---|---|---|---|---|
| | RMSE | MAE | RMSE | MAE | RMSE | MAE |
| MF | 1.2142 | 0.9971 | 1.2928 | 0.9879 | 1.3148 | 1.0066 |
| AFM | 1.1762 | 0.8723 | 1.3006 | 1.1018 | 1.2763 | 0.9724 |
| RecAgent | 1.1021 | 0.7632 | 1.2587 | 1.1191 | 1.0766 | 0.9598 |
| RecMind-SI (few-shot) | 1.0651 | 0.6731 | 1.2139 | 0.9434 | 0.9291 | 0.6981 |
| Agent4Rec | 0.7612 | 0.7143 | 0.8788 | 0.6712 | 0.7577 | 0.6880 |
| SimUSER(sim · persona) | **0.5341** | **0.4671** | **0.5919** | **0.4562** | **0.6153** | **0.5686** |
| SimUSER(zero · w/o persona) | 0.6872 | 0.5981 | 0.7103 | 0.6667 | 0.7089 | 0.6928 |
| SimUSER(zero · persona) | 0.6112 | 0.5353 | 0.6698 | 0.5597 | 0.6844 | 0.6392 |
| SimUSER(sim · w/o persona) | 0.6012 | 0.5421 | 0.6811 | 0.5605 | 0.6912 | 0.6420 |

the model (SimUSER(sim)). This is because the grounded interactions augment the context during decision-making. For instance, when an agent was asked to rate the movie *The City of Lost Children*, it retrieved movies such as *Johnny Mnemonic* and *Judge Dredd*, all featuring similar adventure and sci-fi themes that it had previously enjoyed. In the absence of simulation, the agent relied solely on its (limited) viewing history from the initial dataset. This demonstrates that agents can refine their own preferences for unrated items through interactions with the simulator.

## 4.4 RATING DISTRIBUTION

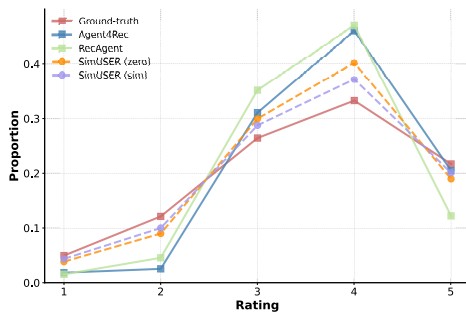

Figure 2: Comparison of rating distributions between ground-truth and human proxies.

Table 3: Evaluation of recommendation strategies on a recommendation task from the MovieLens dataset.

| | $\overline{P}_{\text{view}}$ | $\overline{N}_{\text{like}}$ | $\overline{P}_{\text{like}}$ | $\overline{N}_{\text{exit}}$ | $\overline{S}_{\text{sat}}$ |
|---|---|---|---|---|---|
| Random | 0.298 | 3.17 | 0.248 | 2.87 | 2.70 |
| Pop | 0.401 | 4.12 | 0.379 | 2.89 | 3.35 |
| MF | 0.469 | **5.98** | 0.446 | 3.09 | 3.67 |
| MultVAE | 0.521 | 5.44 | 0.458 | 3.21 | 3.82 |
| LightGCN | **0.552** | 5.50 | **0.451** | **3.34** | **3.97** |

While aligning individual agent ratings with their human counterparts is crucial, it is also necessary that human proxies replicate real-world user behavior at a macro level. This implies ensuring that the distribution of ratings generated by the agents aligns closely with the distributions observed in the original dataset. Figure 2 presents the rating distribution from the MovieLens-1M dataset and the ratings generated by the agents. These results reveal a high degree of alignment between the simulated and actual rating distributions, with a predominant number of ratings at 4 and a small number of low ratings (1-2). While Agent4Rec assigns fewer 1-2 ratings compared to real users, our approach, by retrieving past interactions from the episodic memory, allows agents to contextualize their ratings based on a broader and more consistent understanding of their own preferences. Besides, SimUSER features a more more granular representation of users by incorporating detailed personas (e.g., summary of tastes, personality traits), which yields realistic macro-level behaviors.

## 4.5 RECOMMENDER SYSTEM EVALUATION

Understanding the efficacy of various recommendation algorithms is crucial for enhancing user satisfaction. By simulating human proxies, we can better predict how users will engage with recommender systems, thereby providing valuable interactive metrics. We also conjecture that users experience higher satisfaction with advanced recommendation strategies compared to random rec-

Table 4: Human-likeness score evaluated by GPT-4o in different recommendation domains. Asterisks (*) denote statistically significant improvements over the best baseline (t-test at $p < 0.05$).

|  | MovieLens | AmazonBook | Steam |
|---|---|---|---|
| RecAgent | 3.01 ± 0.14 | 3.14 ± 0.13 | 2.96 ± 0.17 |
| Agent4Rec | 3.04 ± 0.12 | 3.21 ± 0.14 | 3.09 ± 0.16 |
| SimUSER(w/o persona) | 3.60 ± 0.17* | 3.54 ± 0.19* | 3.58 ± 0.26* |
| SimUSER(persona) | **4.22±0.17*** | **3.98±0.18*** | **3.97±0.24*** |

ommendations. We tested several recommendation strategies within our simulator, including most popular (Pop), matrix factorization (MF) Koren et al. (2009), LightGCN He et al. (2020), and Mult-VAE Liang et al. (2018), using the MovieLens dataset. Upon exiting, agents rated their satisfaction with the recommendation system on a scale from 1 to 10. Ratings above 3 were considered indicative of a *like*. The metrics collected included average viewing ratio ($\overline{P}_{\text{view}}$), average number of likes ($\overline{N}_{\text{like}}$), average ratio of likes ($\overline{P}_{\text{like}}$), average exit page number ($\overline{N}_{\text{exit}}$), and average user satisfaction score ($\overline{S}_{\text{sat}}$). Table 3 summarizes the performance metrics for the various recommendation strategies. Agents exhibit higher satisfaction with advanced recommendations compared to random and popularity-based methods, consistent with observed real-life trends. LightGCN outperformed both MF and MultVAE across multiple metrics, confirming its effectiveness in providing personalized recommendations. Namely, LightGCN achieved the highest $P_{\text{view}}$ and $S_{\text{sat}}$ scores, indicating superior user engagement and satisfaction. Overall, these findings highlight the potential of generative agents as a cost-effective technique for simulated A/B testing.

## 4.6 LLM EVALUATOR

LLM Evaluators Chiang & Lee (2023) have validated LLM as an evaluator, achieving comparable performance with expert evaluators. Thereby, we use GPT-4o to evaluate the interactions and feedback generated by generative agents. That is, we collected the interactions of the agents and asked GPT-4o to differentiate whether these interactions are from humans or AI-generated. A 5-point Likert scale questionnaire was used. In this scale, a higher score indicates a closer resemblance to human-like responses. Although both methods leverage the semantic awareness of LLMs to achieve near-human-level performance, the results in Table 4 demonstrate that our method significantly outperforms Agent4Rec. The memory and persona modules are among the main factors contributing to the faithfulness of our method. We also noticed that letting the agent estimate its tiredness, feeling and emotion greatly enhances the believability and consistency of its responses. On the other hand, in Agent4Rec, tendencies to [EXIT] the recommender system early and provide inconsistent ratings for similar items — ranging from low to high, contribute to suspicions of AI involvement.

## 5 CONCLUSION

In this study, we present a simulation framework for leveraging LLMs as believable user proxies. Our two-phase approach includes persona matching and interactive recommender system assessment, seeking to align user interactions more closely with real-world user behaviors. Key novelties of our method include the integration of detailed user personas, a knowledge-graph memory module, alongside integrating visual cues to emulate visual-driven reasoning. The present study also introduces a post-interaction reflection mechanism to let the agent discover high-level insights from interactions. We evaluate SimUSER across various recommendation domains, including movies, books, and video games. Results demonstrate closer alignment of our agents with their human counterparts at both micro and macro levels. We further explore the influence of thumbnails on user engagement and the significance of reviews in user decision-making. Experimental findings highlight the potential of LLM-driven simulations in bridging the gap between offline metrics and online evaluation, paving the way for advancements in the development of systematic and cost-efficient evaluation of recommender systems.

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

## A APPENDIX

**Experimental Settings.** We separate the dataset into training, validation, and test sets (80/10/10%). Relationships between users and items from the training/validation and test sets were excluded from the knowledge graph memory to prevent data leakage. In this paper, we report results for SimUSER with simulation **SimUSER(sim)**, and without simulation **SimUSER(zero)**. In SimUSER(zero), the agent's memory module is initialized from the history of its human counterpart. When the review score for an item is greater than 4, the agent stores a memory entry in the form `I liked {item_name} based on my review score of {score}`. For a score of 2 or below, the following format is utilized `I disliked {item_name} based on my review score of {score}`. Neutral scores result in the entry `I felt neutral about {item_name} based on my review score of {score}`. In SimUSER(sim), agents can also interact with the recommender systems (training set) for up to 20 pages or exit the system at any time. The corresponding interactions are used to enhance the memory module. In all the experiments, items rated $\geq 4$ are considered as liked by the user, while items $\leq 2$ are considered as disliked. These interactions are stored both as plain text in the episodic memory and as relationships in the knowledge graph memory. These simulated interactions with the RS are stored in the episodic memory with the following format: `The recommender system recommended the following {item_type} to me on page {page_number}: {name_all_items}, among them, I selected {watched_items} and rate them {ratings} respectively. I dislike the rest {item_type} items: {dislike_items}`.

In some sets of experiments, we report performance without persona matching SimUSER(w/o persona), and with persona matching SimUSER(persona). In the absence of persona matching, personality traits, age, occupation and taste summary are omitted from the prompts. Matrix factorization (MF) is utilized as the recommender model unless specified otherwise. In our simulator, agents are presented with four items $n = 4$ per page and allowed to interact by viewing and rating items based on their preferences. When the output of the LLM deviated from the desired format, resulting in errors, the LLM was re-prompted with the following instruction: `You have one more chance to provide the correct answer`.

As mentioned above, we leverage `GPT-4o-mini` as the LLM backbone in all the experiments unless stated differently. We use $\alpha = 0.8$ to balance item-item similarity with user-item similarity. We set $k_2 = 3$ when retrieving similar items from the knowledge graph-memory, and $k_1 = 5$ for the episodic memory. The titles and ratings of retrieved items from the knowledge graph are concatenated to condition decision-making prompts. Empirically, we set the weight of node embeddings to 0.25 when computing top-$k_2$ scores. Documents and embedding of text (**E**) were obtained using `text-embedding-3-small`. Given the average rating $\bar{R}$ of a user: $\bar{R} = \frac{1}{N} \sum_{i=1}^{N} r_{ui}$, the pickiness level $P(\bar{R})$ of a user was determined based on the following thresholds:

$$P(\bar{R}) = \begin{cases} P_1 & \text{if } \bar{R} \geq 4.5 \\ P_2 & \text{if } 3.5 \leq \bar{R} < 4.5 \\ P_3 & \text{if } \bar{R} < 3.5 \end{cases}$$

where $P_1$ corresponds to *not picky*, $P_2$ corresponds to *moderately picky*, and $P_3$ corresponds to *extremely picky*.

In Appendix F.6, we compare the results of SimUSER taking as input: 1) the original movie poster, 2) a random screenshot from the movie trailer on YouTube, 3) the original movie poster distorted with a blue color filter (hue=30, lightness=30, saturation=30).

An illustration of the method is provided in Figure 3, detailing the interaction between its components and their roles within the proposed framework.

## B ADDITIONAL IMPLEMENTATION DETAILS

**Persona Matching** 10 candidate personas are generated and used as prior for self-consistency check. To facilitate cost-efficient persona identification, up to 50 items are randomly sampled from the user's viewing history and passed to query the LLM. For simplicity, consistency check also utilizes

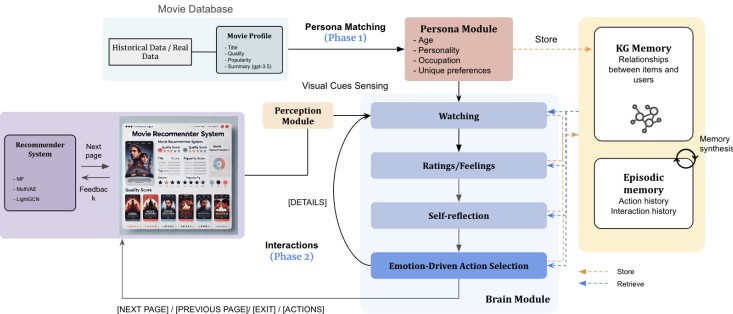

Figure 3: The SimUSER framework for evaluating a movie recommender system.

$\varrho = 50$ items and $j = 3$ for other users $\bar{u}$. The $j = 3$ subsets of the user's interactions are compared with those of 30 other users, which are randomly selected. The following occupations were introduced in the persona-matching prompt in order to enhance diversity:

| Category | Values |
|---|---|
| Occupation | |
| | Other |
| | Academic/educator |
| | Artist |
| | Clerical/admin |
| | College/grad student |
| | Customer service |
| | Doctor/health care |
| | Executive/managerial |
| | Farmer |
| | Homemaker |
| | K-12 student |
| | Lawyer |
| | Programmer |
| | Retired |
| | Sales/marketing |
| | Scientist |
| | Self-employed |
| | Technician/engineer |
| | Tradesman/craftsman |
| | Unemployed |
| | Writer |

We defined the following age categories:

| Category | Values |
|---|---|
| Age | |
| 1 | Under 18 |
| 18 | 18-24 |
| 25 | 25-34 |
| 35 | 35-44 |
| 45 | 45-49 |
| 50 | 50-55 |
| 56 | 56+ |

For personality features, we listed the five Big Five personality traits and their possible ranges:

| Trait | Range |
|---|---|
| Openness | (1-3) |
| Conscientiousness | (1-3) |
| Extraversion | (1-3) |
| Agreeableness | (1-3) |
| Neuroticism | (1-3) |

The score of each big-5 facet was then mapped to the corresponding text in: low, medium, high.

## C  PROMPTS

In this section we describe the prompts used to queried the LLM, with those marked by blue headers constituting the core framework of SimUSER. Prompts highlighted in purple are task-specific prompts utilized in some of the experiments discussed in this paper. Note that for books (Amazon-Book) and video games (Steam), **watching / watch** were respectively replaced with **reading / read** and **playing / play**. For the sake of generality and readability, we report only the **watching/watch** version.

### C.1  PERSONA MATCHING

The following prompt was used to generate personas from historical data:

---

**Persona Generation Prompt**

You are an AI assistant specializing in analyzing human preferences and understanding personas. You will analyze the items liked and disliked by an individual.

The task is as follows: I will provide you with a list of items that the individual liked and disliked. Based on this information, you need to predict the most suitable persona for this individual. The persona includes the individual's occupation, age range, and Big5 personality traits (each scored from 1 to 3, where 1 is low, 2 is medium, and 3 is high).

** Liked Items: **
1. {item 1}
2. {item 2}
...

** Disliked Items: **
1. {item 1}
2. {item 2}
...

Based on the preferences listed above, predict the most suitable persona for this individual. Ensure that the predicted persona is coherent with the liked and disliked items, accurately reflecting the individual's tastes and preferences.

Response Format:
1. **Occupation:** [choose from <occupation list>]
2. **Age Range:** [choose from <age list>]
3. **Big5 Personality Traits:**
- **Openness:** [score from 1 to 3]
- **Conscientiousness:** [score from 1 to 3]
- **Extraversion:** [score from 1 to 3]

---

> - **Agreeableness:** [score from 1 to 3]
> - **Neuroticism:** [score from 1 to 3]
>
> Please provide your prediction based on the provided liked and disliked items, ensuring the persona is coherent with the individual's tastes and preferences.

Scores of the candidate personas were estimated as:

---

**Persona-Matching Prompt**

Pretend to be a {candidate_persona}. Here is a list of {item_type} you like and dislike:

** Liked Items: **
1. {item 1}
2. {item 2}
...

** Disliked Items: **
1. {item 1}
2. {item 2}
...

Here is a summary of your tastes: {taste}
On a scale of 1 to 10, where 1 is the least likely and 10 is the most likely, rate how likely you are to like and dislike similar {item_type} in the future, based on your own taste. Explain the reason for your rating.

**### Response Format:**
Rating:
Reason:

---

where {item_type} refers to the item type, specifically *movie*, *game*, *book* for MovieLens, Steam, and AmazonBook, respectively.

## C.2 TASTE SUMMARY PROMPT

The summary of the user's preferences, {taste}, was obtained by querying the LLM with this prompt:

**Taste Summary Prompt**

Based on the following lists of {item_type} the user liked and disliked, generate a comprehensive summary of the user's taste, preferences, and aversions.

** Liked {item_type}:**
1. {item 1}
2. {item 2}
...

** Disliked {item_type}:**
1. {item 1}
2. {item 2}
...

### Instructions:
1. Analyze the genres, themes, styles, and any other relevant features of the liked {item_type}.
2. Contrast these with the features of the disliked {item_type}.
3. Summarize the user's overall taste, specific preferences, and aversions in a clear and concise manner.

### Response Format:
- Summary of User's Taste: [summary]

## C.3 CAPTION GENERATION

In the implementation of the perception module, a caption is generated for each item using `GPT-4o` as the LLM core, with the following request:

**Caption Generation Prompt**

You are an expert in generating detailed and descriptive captions for images.
You are provided with an image representing a {item_type} poster or a product. Your task is to generate a detailed, descriptive, and concise caption for the image. The caption should accurately describe the visual elements and provide context to help someone decide if they would be interested in clicking on it, as if they were seeing it themselves on a website.

### Data Formats:
1. **Image**: A single image of a {item_type} poster or a product.

### Task:
- Generate a detailed and concise caption that includes: - Visual details: Colors, shapes, objects, actions, and notable features present in the image.
- Contextual information: Scene or item context, genre, mood, characters for {item_type}, or brand, usage, and key features for products.
- Emotional tone: The atmosphere or emotional appeal conveyed by the image.
- Unique selling points: Highlight any unique features that make the item special.
- Clarity and precision: Ensure the description is clear, precise, and accurately represents the image.
- Conciseness: Be detailed yet concise, avoiding overly long descriptions.

### Example:
The movie poster for "Avatar: The Way of Water" features four prominently displayed blue-skinned Na'vi characters with distinct facial expressions, conveying a sense of intensity and determination. The background showcases a breathtaking sunset over a vast ocean, with rocky islands emerging from the water. Below the Na'vi, a dynamic scene captures a Na'vi warrior riding a large, winged creature above the ocean waves, hinting at adventure and freedom. The title "Avatar: The Way of Water" is boldly displayed at the bottom, along with the release date "December 15" The overall mood of the poster is epic and visually stunning, promising an immersive and thrilling cinematic experience set in

> the richly detailed world of Pandora.
>
> Please generate the caption based on the provided image.

where the image is a thumbnail of the item. Thumbnail images were systematically scrapped, using the product name as the request.

## C.4 RETRIEVAL FROM THE EPISODIC MEMORY

When we retrieve documents from the episodic memory, follow-up questions are generated based on the initial {query}, as follows:

---

**Follow-up question Prompt**

**Instruction:**
Given the following query, generate 1-3 follow-up questions that are directly related to the original query. These follow-up questions should help retrieve documents that provide further relevant yet related information on the topic.

**Query:**
{query}

**Follow-up questions:**

---

Note that caching can be employed to reduce the computational footprint of generating follow-up questions

## C.5 WATCHING ITEMS

SimUSER agents rely on the following prompt to decide which items to watch. When interacting with the simulator, the watching and rating/feeling prompts are combined in a chain-of–thoughts fashion as a single query to speedup inference:

---

**Watching Item Prompt**

**### System Prompt**
You excel at role-playing. Picture yourself as a user exploring a {item_type} recommendation system. You have the following habits:

- Your engagement trait is described as: {engagement}
- Your conformity trait is described as: {conformity}
- Your variety trait is described as: {variety}

The engagement characteristic pertains to the frequency of your {item_type}-watching habits. The conformity characteristic measures the degree to which your ratings are influenced by historical ratings. The variety characteristic gauges your likelihood of watching {item_type} that may not align with your usual taste.

Pretend to be a {persona_description}. You are {pickiness} about {item_type}. You only watch {item_type} that align with your taste.

## Recommended List ##
{item_list}

**### Instructions**
1. Review each {item_type} in the ## Recommended List ##.
2. For each {item_type}, decide if it aligns with your taste. If it aligns, decide to watch it and provide a brief reason. If it does not align, provide a brief reason.

---

3. Rate each {item_type} you choose to watch on a scale of 1 to 5 and provide a brief feeling or reaction. If you do not watch the {item_type}, mark the rating as N/A.
4. Follow the exact format for each {item_type} as shown below.

### Format
{item_type}: [item name]; WATCH: [yes or no]; REASON: [brief reason]; RATING: [1 to 5 if watched, N/A if not watched]; FEELING: [brief feeling]

### Example

MOVIE: The Lion King (1994); WATCH: yes; REASON: It's an animated movie with a touch of adventure and fantasy, which aligns with my preferences; RATING: 5; FEELING: It was a magical and emotional journey that I could not resist.

MOVIE: Titanic (1997); WATCH: no; REASON: Does not fit my preference for animated movies or fantasy elements; RATING: N/A; FEELING: N/A

### Your Response

where {item_type} specifies the type of item, such as *movie*, *game*. The placeholders {engagement}, {conformity}, and {variety} denote habits estimated by the persona module. {persona_description} is a plain text description of the user's persona (see below). {pickiness} reflects the user's pickiness regarding the items. Finally, {item_list} comprises the list of items recommended by the system, which comprises similar items retrieved from the knowledge graph memory.

## C.6 RATING ITEM/FEELING

The rating/feeling prompt is derived from the *watching* prompt. The watching prompt was substituted with the following *rating* prompt in experiments that only require to rate the items — when the agents do not have to decide which items to watch:

---

**Rating Item Prompt**

### System Prompt
You excel at role-playing. Picture yourself as a user exploring a {item_type} recommendation system. You have the following habits:

- Your engagement trait is described as: {engagement}
- Your conformity trait is described as: {conformity}
- Your variety trait is described as: {variety}

The engagement characteristic pertains to the frequency of your {item_type}-watching habits. The conformity characteristic measures the degree to which your ratings are influenced by historical ratings. The variety characteristic gauges your likelihood of watching {item_type} that may not align with your usual taste.

Pretend to be a {persona_description}. You are {pickiness} about {item_type}. You only watch {item_type} that align with your taste.

## Recommended List ##
{item_list}

### Instructions
1. Review each {item_type} in the ## Recommended List ##.
2. For each {item_type}, decide if it aligns with your taste.
3. Rate each {item_type} you choose to watch on a scale of 1 to 5 and provide a brief feeling / reaction.

---

> 4. Follow the exact format for each {item_type} as shown below.
>
> **### Format**
> {item_type}: [item name]; RATING: [1 to 5 if watched, N/A if not watched]; FEELING: [brief feeling]
> **### Example**
> MOVIE: The Lion King (1994); RATING: 5; FEELING: It's an animated movie with a touch of adventure and fantasy, which aligns with my preferences. It was a magical and emotional journey that I could not resist.
> MOVIE: Titanic (1997); RATING: 2; FEELING: Not particularly interested in romantic-dramatic themed movies.
> **### Your response**

As done in the **Watching Prompt** (Appendix C.5), this prompt aggregates similar movies retrieved from the KG memory to reduce hallucination and enhance the reasoning context (see below *page format*). Namely, each presented movie is accompanied with a list of similar movies, identified based on relationships between users and items, as well as semantic information.

## C.7 PAGE FORMAT

The pages of the recommender simulator ({item_list}) were formatted as follows:

> **Page Format**
>
> PAGE {page_number}
> <− {item_title} −> <− History ratings: {item_rating} −> <− Summary: {item_description} −> <− Thumbnail: {thumbnail} −> <− Similar items: {similar_items} −>
>
> <− {item_title} −> <− History ratings: {item_rating} −> <− Summary: {item_description} −> <− Thumbnail: {thumbnail} −> <− Similar items: {similar_items} −>
>
> ...

where {page_number} represents the page number in the recommender simulation, {item_title} denotes the title or name of the item, {item_rating} is the average historical rating of the item, and {item_description} provides a description of the item. By default, a short description is displayed, but if the agent decides to click on the item, {item_description} is substituted with a more detailed description.

When available, we leverage short descriptions provided in the original Agent4Rec paper Hou et al. (2024). In other domains, we employ GPT-4o to generate the content description of each item using the following prompt: `Summarize the {item_type} {title} with one sentence. The answer cannot include the {item_type} title.` Detailed descriptions were generated using the following prompt: `Summarize the {item_type} {title} with two to three sentences. The answer cannot include the {item_type} title.` The response is used as the item description.

The {thumbnail} placeholder contains the caption generated by the captioner model based on the item's thumbnail. Note that {thumbnail} is an optional field that can be omitted if the item lacks a thumbnail. {similar_items} lists similar items retrieved from the knowledge graph memory, with `**title (rating/5)**` separated by commas as the format. The `rating` is replaced with the user's individual rating when available, or alternatively, with the average rating of other users stored in the KG memory.

## C.8 PERSONA DESCRIPTION

The {persona_description} that describes in plain text the persona of a user was set as:

> **Persona Format**
>
> {age} user. You are a {occupation}. You have a {openness} openness, {conscientiousness} conscientiousness, {extraversion} extraversion, {agreeableness} agreeableness, and {neuroticism} neuroticism. Summary of your tastes: {taste}.

where the big-5 personality traits were replaced with the corresponding level (low, medium, high), based on the score obtained during the persona matching stage. {taste} represents the user's preferences, summarized by $s_u$ that is obtained during persona matching, see Appendix C.2.

## C.9  POST-INTERACTION REFLECTION

The post-interaction prompt was formatted as:

> **Post-Interaction Prompt**
>
> You are an {item_type} recommendation system. Analyze the user's recent interactions with {item_type} and their feedback to gain insights into their preferences and suggest future content.
>
> **### User Recent Interactions:**
> - Item watched: {item_title}
> - History rating: {item_rating}
> - User feedback: {rating/feeling}
>
> **### Example Salient Questions:**
> - What genres does the user consistently rate highly?
> - Are there specific themes or elements that the user enjoys?
> - How does the user's preference for certain genres compare to their past behavior?
>
> **### Tasks:**
> 1. Identify and list three most salient questions about the user's preferences and behaviors based on the recent interactions.
> 2. Provide three high-level insights about the user's current preferences. Cite the particular records that served as evidence for each insight.
>
> **### Your Response:**

where {rating/feeling} consists of the ratings and feelings expressed by the synthetic user.

## C.10  NEXT ACTION

The next action is selected by querying the large language model with this prompt:

> **Action Prompt**
>
> **### Instructions**
> You have the following habits:
>
> - Your engagement trait is described as: {engagement}
> - Your conformity trait is described as: {conformity}
> - Your variety trait is described as: {variety}
>
> The engagement characteristic pertains to the frequency of your {item_type}-watching habits. The conformity characteristic measures the degree to which your ratings are influenced by historical ratings. The variety characteristic gauges your likelihood of watching {item_type} that may not align with your usual taste.

Pretend to be a {persona_description}. Follow these steps carefully to decide your next action:

1. Generate an overall feeling based on your memory, in accordance with your engagement trait and your satisfaction with the recommender system.

- Evaluate the relevance and interest of each previously shown {item_type}. Consider how well the previous recommendations aligned with your tastes and interests.
- If your overall feeling is positive, write: POSITIVE: [reason].
- If it's negative, write: NEGATIVE: [reason].

2. Assess your level of fatigue. Consider that you may become tired more easily if you have an inactive engagement trait. Indicate your level of tiredness: NOT TIRED, A LITTLE TIRED, VERY TIRED.

3. Assess your current emotion. Reflect on your past interactions to estimate your current emotion. Choose one emotion from the following list that best describes how you feel: CURIOUS, FRUSTRATED, EXCITED, NEUTRAL, OVERWHELMED.

4. Select the next decision:

- Decide whether to exit browsing based on your overall feeling, engagement trait, and tiredness. You will exit the recommender system if you have negative feelings or if you are tired, especially if you have a low engagement trait. To leave, write: [EXIT]; Reason: [brief reason].
- Decide whether to continue browsing based on your overall feeling, engagement trait, and tiredness. To continue browsing, write: [NEXT]; Reason: [brief reason].
- Click on an item to get its full description. To click on an item, write: [CLICK]; Reason: [brief reason].
- Go back to the previous page. To go back, write: [PREVIOUS]; Reason: [brief reason].

### Example 1
1. Overall Feeling: POSITIVE: The recommendations align well with my tastes and the system is easy to use.
2. Tiredness Level: A LITTLE TIRED
3. Emotion: EXCITED
4. Decision: [NEXT]; Reason: I am enjoying the recommendations, and feel that I can continue browsing for a bit longer.

### Example 2
1. Overall Feeling: NEGATIVE: The recommendations do not match my preferences, and I find the system frustrating.
2. Tiredness Level: VERY TIRED
3. Emotion: FRUSTRATED
4. Decision: [EXIT]; Reason: I am dissatisfied with the recommendations and too tired to continue browsing.

### Your Turn
1. Overall Feeling:
2. Tiredness Level:
3. Current Emotion:
4. Decision:

Given the above-mentioned instructions, an agent can choose from the following actions:

- **Exit the recommender system** ([EXIT]): If the agent experiences negative feelings or significant fatigue, they may choose to exit the system.
- **Continue browsing** ([NEXT]): If agents have positive feelings and sufficient energy, they may decide to continue browsing.

- **Click on an item to get its full description** (`[CLICK]`): When agents want more information about a specific item, they may select this option.

- **Go back to the previous page** (`[PREVIOUS]`): If agents wish to return to a previous set of recommendations, they may choose this action.

## C.11 POST-INTERVIEW PROMPT

The prompt presented to each agent for post-interview is as follows:

---

**Post-Interview Prompt**

How satisfied are you with the recommender system you recently interacted with?

**### Instructions:**
1. Rating: Provide a rating from 1 to 10.
2. Explanation: Explain the reason for your rating.

**### Response Format:**
- RATING: [integer between 1 and 10]
- REASON: [detailed explanation]

---

## C.12 BELIEVABLY OF SYNTHETIC USER PROMPT

In Section 4.2, the rating prompt is modified with the following instructions:

---

**Believably of Synthetic User Prompt**

**### Instructions**
1. Review each {item_type} in the ## Recommended List ##.
2. For each {item_type}, classify if you have already interacted with it ("Interacted") or if you have not ("Not Interacted").

---

## C.13 LLM EVALUATOR PROMPT

The prompt below was employed to distinguish between humans and AI-generated interactions:

---

**LLM Evaluator Prompt**

Please evaluate the following interactions of an agent with a recommender system, and determine whether it is generated by a Large Language Model (LLM) AI or a real human: {interaction logs}

Please rate on a scale of 1 to 5, with 1 being most like an AI and 5 being most like a human.

---

# D KNOWLEDGE-GRAPH CONSTRUCTION

The knowledge graph memory is initially populated with historical data from datasets such as MovieLens, Steam, and AmazonBook. Missing information were retrieved from IMDb and/or Wikipedia.

**MovieLens.** The MovieLens dataset provides detailed information about users, movies, and ratings. We use the version (MovieLens-1M) that includes 1 million user ratings. The construction of the knowledge graph from this dataset involves creating nodes for users and movies and defining edges based on their interactions and attributes. Namely, it incorporates various entity types as nodes:

User, Movie, Genre, Director, Actor, Rating, Release Year, Occupation, and Age. The knowledge graph contains edges that depict relationships between the following relationships:

- Interaction edges: rated-by, rated, viewed-by, viewed, given-to, gives-rating,
- Preference edges: liked-by, liked, disliked-by, disliked,
- Attribute edges: has-occupation, released-in, contains, has-users, belongs-to, directed-by, directed, features, acts-in, has-age, has-users.

**Steam.** The Steam dataset comprises 2,567,538 users, 15,474 games, and 7,793,069 English reviews crawled from Steam, a large video game distribution platform. The dataset also includes rich information that might be useful in future work, like users' play hours, pricing information, media score, category, developer (etc.). The construction of the knowledge graph from this dataset involves creating nodes for users and games and defining edges based on their interactions and attributes. The knowledge-graph memory for the Steam network was created as depicted below. The knowledge graph incorporates various entity types as nodes: User, Game, Genre, Developer, Publisher, Tag, Review Sentiment, Price, Release Date, Age, and Occupation. The knowledge graph contains edges that depict relationships between the following relationships:

- Interaction edges: reviewed-by, reviewed, viewed-by, viewed, purchased-by, purchased,
- Preference edges: liked-by, liked, disliked-by, disliked,
- Attribute edges: developed-by, published-by, belongs-to-genre, tagged-as, has-review-sentiment, has-price, released-on-date, has-age, has-occupation.

**AmazonBook.** The AmazonBook dataset provides detailed information about users, books, and reviews. We use the dataset that includes reviews and book details. The construction of the knowledge graph from this dataset involves creating nodes for users and books and defining edges based on their interactions and attributes. The knowledge graph incorporates various entity types as nodes: User Nodes, Book Nodes, Genre, Author, Publisher, Tag, Review Sentiment, Price, Release Date, Age, and Occupation. The knowledge graph contains edges that depict relationships between the following relationships:

- Interaction edges: reviewed-by, reviewed, viewed-by, viewed, purchased-by, purchased,
- Preference edges: liked-by, liked, disliked-by, disliked,
- Attribute edges: belongs-to-genre, written-by, published-by, tagged-as, has-review-sentiment, has-price, released-on-date, has-age, has-occupation,
- User-tag edges: tags, tagged-by,
- Also bought and also viewed edges: also-bought-by, also-bought, also-viewed-by, also-viewed,
- Bought together edges: bought-by, bought-together.

**Graph-Aware Retrieval.** To balance computational efficiency and the richness of the captured relationships, we set the maximum length of the relationship-paths to 4. Here are some examples of meta-paths for the MovieLens dataset that characterize user-user relationships:

- **Movie-User-Movie Rate**: Movie $\xrightarrow{\text{rated-by}}$ User $\xrightarrow{\text{rated}}$ Movie
- **Movie-User-Movie Viewed**: Movie $\xrightarrow{\text{viewed-by}}$ User $\xrightarrow{\text{viewed}}$ Movie
- **Movie-User-Movie Liked**: Movie $\xrightarrow{\text{liked-by}}$ User $\xrightarrow{\text{liked}}$ Movie
- **Movie-User-Movie Disliked**: Movie $\xrightarrow{\text{disliked-by}}$ User $\xrightarrow{\text{disliked}}$ Movie
- **Movie-Genre-Movie**: Movie $\xrightarrow{\text{belongs-to}}$ Genre $\xrightarrow{\text{contains}}$ Movie
- **Movie-Director-Movie**: Movie $\xrightarrow{\text{directed-by}}$ Director $\xrightarrow{\text{directed}}$ Movie
- **Movie-Actor-Movie**: Movie $\xrightarrow{\text{features}}$ Actor $\xrightarrow{\text{acts-in}}$ Movie
- **Movie-User-Rating-Movie**: Movie $\xrightarrow{\text{rated-by}}$ User $\xrightarrow{\text{gives-rating}}$ Rating $\xrightarrow{\text{given-to}}$ Movie

- **Movie-ReleaseYear-Movie**: Movie $\xrightarrow{\text{released-in}}$ ReleaseYear $\xrightarrow{\text{contains}}$ Movie
- **Movie-Occupation-Movie**: Movie $\xrightarrow{\text{rated-by}}$ User $\xrightarrow{\text{has-occupation}}$ Occupation $\xrightarrow{\text{has-users}}$ User $\xrightarrow{\text{rated}}$ Movie
- **Movie-Age-Movie**: Movie $\xrightarrow{\text{rated-by}}$ User $\xrightarrow{\text{has-age}}$ Age $\xrightarrow{\text{has-users}}$ User $\xrightarrow{\text{rated}}$ Movie

**Steam.** Here are some examples of meta-paths for the Steam dataset that characterize relationships in the KG:

- **Game-User-Game Review**: Game $\xrightarrow{\text{reviewed-by}}$ User $\xrightarrow{\text{reviewed}}$ Game
- **Game-User-Game Viewed**: Game $\xrightarrow{\text{viewed-by}}$ User $\xrightarrow{\text{viewed}}$ Game
- **Game-User-Game Liked**: Game $\xrightarrow{\text{liked-by}}$ User $\xrightarrow{\text{liked}}$ Game
- **Game-User-Game Disliked**: Game $\xrightarrow{\text{disliked-by}}$ User $\xrightarrow{\text{disliked}}$ Game
- **Game-Genre-Game**: Game $\xrightarrow{\text{belongs-to-genre}}$ Genre $\xrightarrow{\text{contains}}$ Game
- **Game-Developer-Game**: Game $\xrightarrow{\text{developed-by}}$ Developer $\xrightarrow{\text{developed}}$ Game
- **Game-Publisher-Game**: Game $\xrightarrow{\text{published-by}}$ Publisher $\xrightarrow{\text{published}}$ Game
- **Game-User-ReviewSentiment-Game**: Game $\xrightarrow{\text{reviewed-by}}$ User $\xrightarrow{\text{has-review-sentiment}}$ ReviewSentiment $\xrightarrow{\text{reviewed}}$ Game
- **Game-Price-Game**: Game $\xrightarrow{\text{has-price}}$ Price $\xrightarrow{\text{contains}}$ Game
- **Game-ReleaseDate-Game**: Game $\xrightarrow{\text{released-on-date}}$ ReleaseDate $\xrightarrow{\text{contains}}$ Game
- **Game-Tag-Game**: Game $\xrightarrow{\text{tagged-as}}$ Tag $\xrightarrow{\text{contains}}$ Game
- **Game-Occupation-Game**: Game $\xrightarrow{\text{reviewed-by}}$ User $\xrightarrow{\text{has-occupation}}$ Occupation $\xrightarrow{\text{has-users}}$ User $\xrightarrow{\text{reviewed}}$ Game
- **Game-Age-Game**: Game $\xrightarrow{\text{reviewed-by}}$ User $\xrightarrow{\text{has-age}}$ Age $\xrightarrow{\text{has-users}}$ User $\xrightarrow{\text{reviewed}}$ Game

**AmazonBook.** The following relationship paths may be considered when retrieving from the AmazonBook network:

- **Book-User-Book Review**: Book $\xrightarrow{\text{reviewed-by}}$ User $\xrightarrow{\text{reviewed}}$ Book
- **Book-User-Book Viewed**: Book $\xrightarrow{\text{viewed-by}}$ User $\xrightarrow{\text{viewed}}$ Book
- **Book-User-Book Liked**: Book $\xrightarrow{\text{liked-by}}$ User $\xrightarrow{\text{liked}}$ Book
- **Book-User-Book Disliked**: Book $\xrightarrow{\text{disliked-by}}$ User $\xrightarrow{\text{disliked}}$ Book
- **Book-Genre-Book**: Book $\xrightarrow{\text{belongs-to-genre}}$ Genre $\xrightarrow{\text{contains}}$ Book
- **Book-Author-Book**: Book $\xrightarrow{\text{written-by}}$ Author $\xrightarrow{\text{wrote}}$ Book
- **Book-Publisher-Book**: Book $\xrightarrow{\text{published-by}}$ Publisher $\xrightarrow{\text{published}}$ Book
- **Book-User-ReviewSentiment-Book**: Book $\xrightarrow{\text{reviewed-by}}$ User $\xrightarrow{\text{has-review-sentiment}}$ ReviewSentiment $\xrightarrow{\text{reviewed}}$ Book
- **Book-Price-Book**: Book $\xrightarrow{\text{has-price}}$ Price $\xrightarrow{\text{contains}}$ Book
- **Book-ReleaseDate-Book**: Book $\xrightarrow{\text{released-on-date}}$ ReleaseDate $\xrightarrow{\text{contains}}$ Book
- **Book-Tag-Book**: Book $\xrightarrow{\text{tagged-as}}$ Tag $\xrightarrow{\text{contains}}$ Book
- **Book-User-Book Bought Together**: Book $\xrightarrow{\text{bought-by}}$ User $\xrightarrow{\text{bought-together}}$ Book

- **Book-User-Book Also Bought**: Book $\xrightarrow{\text{also-bought-by}}$ User $\xrightarrow{\text{also-bought}}$ Book

- **Book-User-Book Also Viewed**: Book $\xrightarrow{\text{also-viewed-by}}$ User $\xrightarrow{\text{also-viewed}}$ Book

- **Book-Occupation-Book**: Book $\xrightarrow{\text{reviewed-by}}$ User $\xrightarrow{\text{has-occupation}}$ Occupation $\xrightarrow{\text{has-users}}$ User $\xrightarrow{\text{reviewed}}$ Book

- **Book-Age-Book**: Book $\xrightarrow{\text{reviewed-by}}$ User $\xrightarrow{\text{has-age}}$ Age $\xrightarrow{\text{has-users}}$ User $\xrightarrow{\text{reviewed}}$ Book

# E  PSEUDO-CODE

We present the pseudo-code for SimUSER agent.

---

**Algorithm 1** SimUSER Algorithm

---
1: **Input:** Historical data $H_u$ for user $u$
2: **Output:** Simulated interactions and feedback
3: **Phase 1: Persona Matching**
4: $\mathcal{P} \leftarrow$ Generate persona from $H_u$
5: $p \leftarrow$ Identify best persona $\in \mathcal{P}$ using self-consistency score
6: **Phase 2: Simulate Interactions**
7: Initialize memory module from $H_u$
8: **repeat**
9:     Perceive the page and items                    ▷ Generate captions
10:     Retrieve similar items from the KG memory
11:     Decide what items to watch
12:     Rate the items and provide feelings
13:     Decide next action $a$ based on satisfaction, fatigue, and emotion
14:     Perform post-interaction reflection
15:     Update memory module
16:     **if** $a = $ [EXIT] **then**
17:         **break**
18:     **else**
19:         Perform action $a$
20: **until** Maximum number of pages reached
21: **Return** Simulated interactions, metrics, and feedback

---

# F  ADDITIONAL EXPERIMENTS

## F.1  ALIGNMENT: RATING VS FEELING

Expressing aligned reviews and ratings is of primary importance to simulate realistic human proxies. Thus, in this section we delve into the alignment between ratings and sentiments. In detail, we prompt the agent to assume one has interacted with a certain item, and ask about its rating and feelings on it. Reviews and ratings from IMDB Maas et al. (2011) are used as ground truth since MovieLens does not contain reviews. After getting a collection of responses, we conduct sentiment-based analysis with PyABSA Yang et al. (2023b). We compare the rating and sentiment distributions of: humans, RecAgent, Agent4Rec, and SimUSER. As depicted in Figure 4, our agents generate ratings aligned with their opinions. For instance, ratings $\geq 4$ are typically associated with positive sentiments. In contrast, Agent4Rec exhibits a bias towards positive opinions, resulting in more positive feelings about the items, including when generating low ratings. It is noteworthy that SimUSER agents and genuine humans express similar sentiments at a macro level.

## F.2  PREFERENCE COHERENCE

Under this scenario, we aim to evaluate whether agents prefer positive recommendations based on a query. Namely, for each request in the Reddit dataset He et al. (2023), we sample: (1) a comment

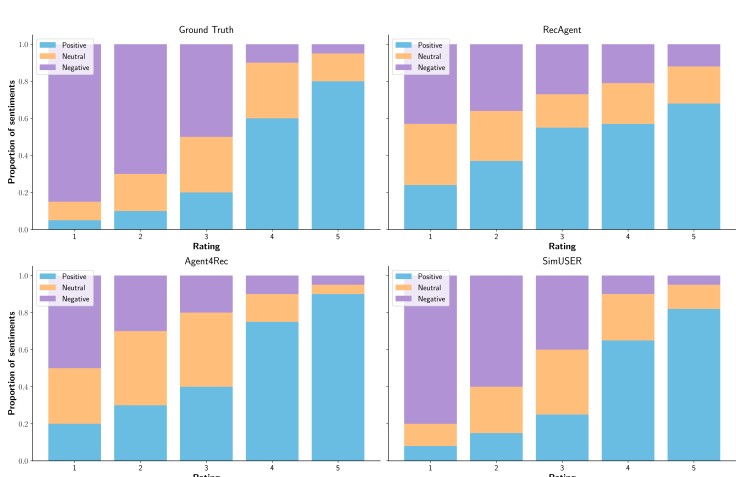

Figure 4: Ratings vs feelings on IMDB dataset. Comparison between human (top left) and LLM-empowered agents.

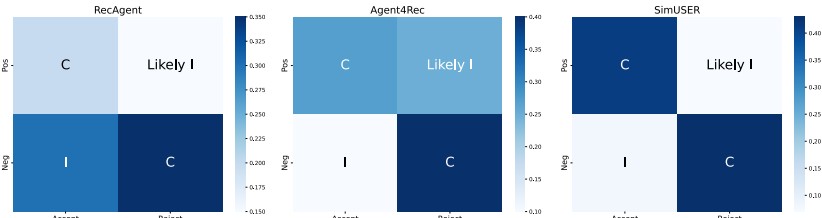

Figure 5: Preference coherence (accept/reject task). 'I' stands for incoherent; 'C' stands for coherent.

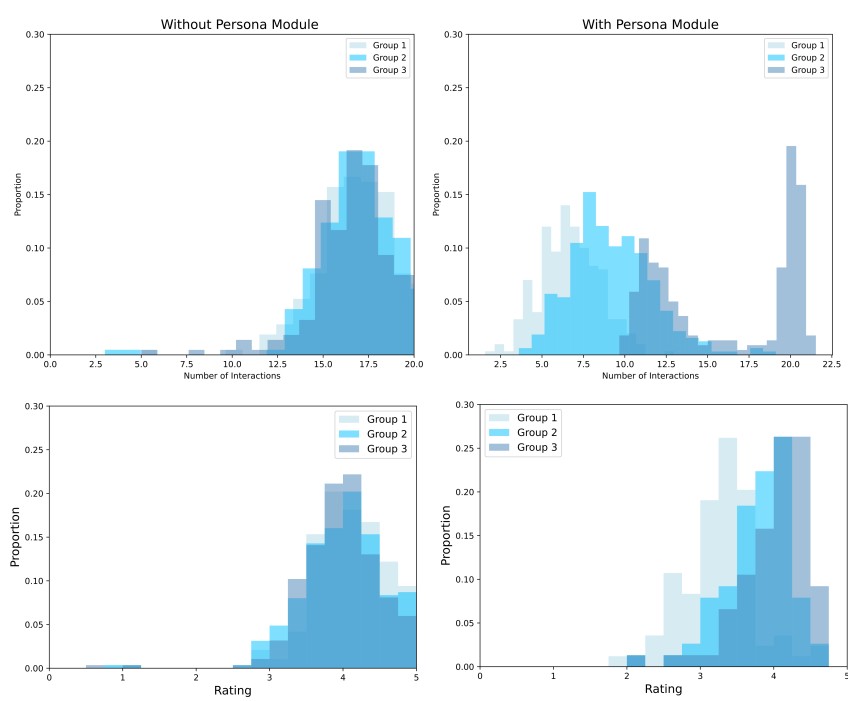

Figure 6: Distribution of interaction numbers (top) and average ratings (bottom) for 3 groups of personas. The left column does not use persona module while the right employs a persona module.

from this request (positive recommendation) (2) a random comment (negative recommendation). The agent is then prompted to decide which items to *watch*. Ideally, synthetic users should watch only positive recommendations and decline negative ones. Behavior is incoherent when the simulator accepts a negative recommendation. We clearly see in Figure 5 that our agents are overall coherent, but sometimes prefer negative recommendations, its proportion being around 4%. Particularly, Agent4Rec agents often accept recommendations that are not aligned with their age and personality. Upon looking at the responses, we notice that providing explanations, as stated in our prompts, reduces incoherent behaviors in synthetic humans.

### F.3 Impact of Persona on User Behaviors

In real life, user behaviors are driven by factors beyond mere individual tastes, including personality traits and demographic attributes such as age, and occupation. To account for these factors, we developed a persona module that incorporates these characteristics. Using the MovieLens dataset, agents were categorized based on their age, occupation (student, office worker, other), and personality traits — categorical features were encoded using one-hot representation, via k-means (k=3). We hypothesize that agents with similar preferences may still exhibit distinct behavior patterns due to their unique persona. Figure 6 depicts the distribution of interaction numbers and average ratings among agents with different personas.

As shown in the figure, it is evident that agents from different groups demonstrate distinct behaviors. We particularly noticed a higher average rating in group 2 and group 3 than in group 1. Instead, synthetic users without personas display more homogeneous behaviors. Upon looking at the users in each group, our findings indicate that younger users (predominantly in group 3, avg age=19.6) tend to engage more frequently with the system, showing higher interaction numbers compared to older users (group 1, avg age=40.2). This increased activity is likely due to their higher adaptability

and interest in exploring new recommendations. In contrast, *older users* exhibit fewer interactions, possibly due to ingrained preferences and less willingness to engage with new content.

### F.4    POST-INTERACTION INTERVIEWS

---

**Case Study of Satisfaction Interview**

**Question:** How satisfied are you with the recommender system? Please rate your satisfaction on a scale from 1 to 10 and provide an explanation for your rating.

**Rating:** 7

**Reason:** As a 35-year-old film enthusiast with a preference for independent films and documentaries, I found the recommender system fairly effective in suggesting thought-provoking content. It recommended several documentaries and indie films that I enjoyed.

However, the system often leaned towards mainstream and popular films rather than exploring unique and lesser-known titles. This tendency occasionally led to less satisfying recommendations, as I value discovering independent, high-quality films.

Additionally, while the system recognized my interest in dramas and thrillers, it missed specific sub-genres like neo-noir and psychological thrillers that I particularly enjoy. Improving the system's ability to identify these nuanced preferences could enhance its effectiveness.

Overall, the system shows strengths in general recommendations but would benefit from a better balance between popular and niche films.

---

Figure 7: One case study of post-exit feeling interview. The blue color highlights the user's satisfaction score.

A central advantage of LLM-based evaluation is their capacity to generate human-comprehensible explanations. Such explanations provide valuable insights that can be employed to refine the recommender system. Figure 7 illustrates a post-interaction interview scenario, where the agent rates its experience with the recommender and returns an explanation based on its personal tastes, habits, emotional memory, and interaction history. The agent noticed that the recommender system generally aligned well with its tastes, particularly recommending movies suitable for its age and personality. However, some factors reduce the overall satisfaction. For instance, despite the agent's preference for sub-genres like neo-noir and psychological thrillers such as "The Sixth Sense" and "Apt Pupil", the system frequently recommended mainstream blockbusters like "Star Wars: Episode I", "The Phantom Menace". These findings underscore the importance of understanding the nuanced behavior of different recommendation algorithms and their suitability for each group of users.

### F.5    RATING ITEMS UNDER HALLUCINATION

In this section, we specifically target items that are unfamiliar to the LLM, seeking to evaluate the ability of SimUSER to mitigate hallucination through its memory module. Similarly to Section 4.3, users are asked to rate movies (MovieLens). Nevertheless, we exclusively include items that are detected as unknown to the LLM. These items $i$ are identified by querying the LLM to classify each movie into one of 18 genres. If the LLM's genre classification matches the actual category $g_i$, it indicates that the LLM is familiar with the item, and such movies are excluded from the experiment. From Figure 8, it is evident that while the RMSE values for all methods increase under hallucination, the performance degradation of SimUSER is less severe compared to the baselines. This relative robustness of SimUSER can be attributed to its KG memory, which effectively mitigates the impact of hallucination by leveraging relationships between users/movies/ratings from previous interactions. By comparing the unfamiliar movie with these similar, well-known ones, the LLM can anchor its predictions in familiar contexts, reducing the likelihood of hallucinations and leading to more accurate ratings. Furthermore, beyond mere text-based descriptions, the item captions provide an additional signal when deciding which items the user likes.

### F.6    THUMBNAIL QUALITY EFFECT

Emotions largely shape decision-making in the recommendation domain. At the center of emotion, images are powerful stimuli that motivate our choices. In light of this, a question arises: Can

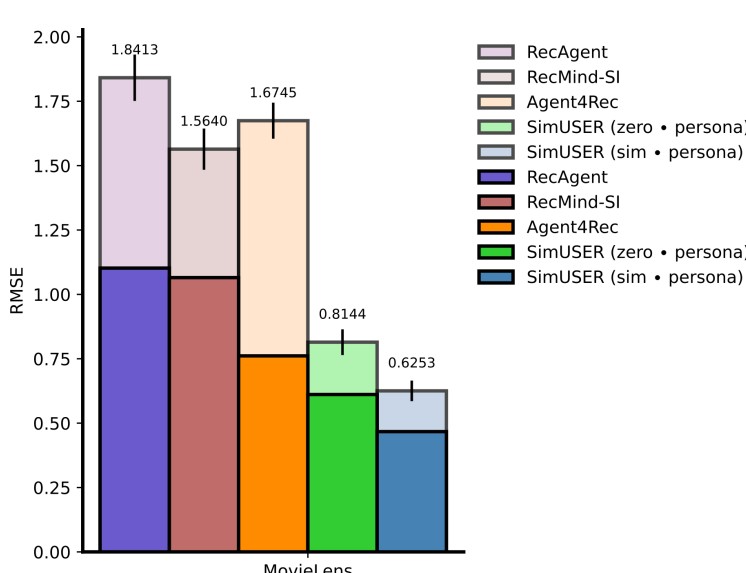

Figure 8: Comparison of RMSE values for original (dark colors) and hallucination-affected (light colors) models for the rating task (MovieLens). The original results are displayed on top of the hallucination results to highlight the performance degradation under hallucination.

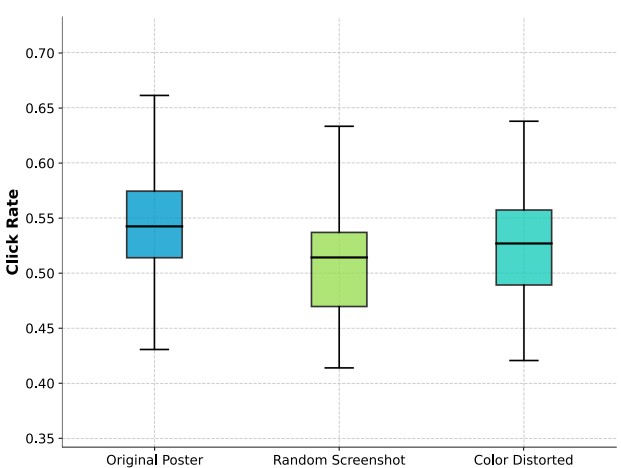

Figure 9: Effect on visual cues on rating distribution for different thumbnail types.

Figure 10: Heatmap showing the impact of biased recommendations on genre ratings over time — exposure effect. The genres and their ratings are displayed after 5, 20, and 50 pages scrolled.

SimUSER be useful in assessing the quality of items' thumbnails? To understand the factors influencing ratings, we randomly selected 100 movies from the MovieLens dataset and ask 100 agents whether they want to watch it. For each movie, we collected three versions of the thumbnails: 1) the original movie poster, 2) a random screenshot from the movie trailer on YouTube, and 3) the original movie poster distorted with a blue color filter. Based on the click rates shown in Figure 9, we notice that high-quality thumbnails — original posters, significantly influence users' inclination to watch a movie. Specifically, original posters lead to higher engagement compared to random screenshots and color-distorted posters. This result highlights SimUSER's capability to reflect the quality of item images in decision-making processes, mirroring trends commonly observed in real-world recommender systems.

## F.7    EXPOSURE EFFECT IN RECOMMENDATION

To assess how biased recommendations shape user preferences over time, we introduce a scenario where the recommender system only recommends two movie categories: *action* and *horror*. It emulates an exposure effect Färber et al. (2023), where repeated exposures to a particular stimulus increase an individual's preference for that stimulus. In the context of recommender systems, repeated exposure to specific genres could amplify user preferences for those genres. Under this scenario, we record the average movie ratings for each category after 5, 20, and 50 pages scrolled by the agents. Namely, the 50 agents are prompted to rate 500 randomly selected movies. Figure 10 illustrates a tendency of the agents to rate items of categories that are over-represented higher during the interactions with the recommender system, particularly after more than 20 pages. Conversely, categories that differ significantly from *action* and *horror* genres generally tend to receive lower average ratings. Experimental results validate SimUSER's capability to replicate the exposure effect, although further research and validation are required with alternative datasets.

## F.8    USER REVIEW INFLUENCE

User proxies may help researchers in identifying the psychological effect of reviews on human interactions. To investigate this effect, we modified the recommendation simulator to display a) the number of reviews, b) one random negative comment, or c) one random positive comment for each item on the recommendation page. We report in Table 5 the average viewing ratio $\overline{P}_{\text{view}}$ and ratio of likes $\overline{P}_{\text{like}}$. We can see that displaying the number of reviews slightly improves the viewing ratio, especially for items having enough reviews (i.e., more than 20 reviews). This aligns with humans' inclination to select popular items in real-life scenarios. On the other, there is no significant

Table 5: Impact of user reviews on recommender System performance.

| Condition | MF | | MultVAE | | LightGCN | |
|---|---|---|---|---|---|---|
| | $\overline{P}_{\text{view}}$ | $\overline{P}_{\text{like}}$ | $\overline{P}_{\text{view}}$ | $\overline{P}_{\text{like}}$ | $\overline{P}_{\text{view}}$ | $\overline{P}_{\text{like}}$ |
| Origin | 0.469 | 0.446 | 0.501 | 0.458 | 0.552 | 0.451 |
| + With # Reviews | 0.481 | 0.479 | 0.537 | 0.470 | 0.549 | 0.486 |
| + With Negative | 0.428 | 0.420 | 0.482 | 0.435 | 0.512 | 0.419 |
| + With Positive | 0.467 | 0.487 | 0.553 | 0.487 | 0.568 | 0.486 |

Table 6: nDCG@k (k=10) and F1-score@k (k=10) for three recommender systems, using either offline or SimUSER-generated interactions.

| Method | nDCG@10 | | F1-score@10 | |
|---|---|---|---|---|
| | Offline | SimUSER | Offline | SimUSER |
| MF | 0.226 | 0.204 | 0.165 | 0.150 |
| MultVAE | 0.288 | 0.257 | 0.180 | 0.172 |
| LightGCN | **0.423** | **0.454** | **0.227** | **0.246** |

difference in $\overline{P}_{\text{like}}$ (t-test $p > 0.05$). Another observation is that displaying negative reviews has a stronger impact on user behavior than showing positive reviews, with a decrease in both the average viewing ratio and number of likes. One possible explanation is that negative reviews discourage users from watching an item, while positive reviews primarily encourage users who are already inclined to watch it to proceed with their choice.

## F.9 SimUSER vs. Offline Metrics

We aim to investigate whether SimUSER can reliably estimate traditional metrics such as nDCG@k (k=10) and F1-score@k (k=10) by comparing the results from traditional offline evaluation with those from SimUSER-generated interactions. For this purpose, we evaluate three recommender systems using the MovieLens dataset under identical conditions for both offline and SimUSER-based evaluations.

Table 6 reports the nDCG@k and F1-score@k values ($k$=10) for both evaluation strategies. In the SimUSER scenario, interactions are generated by our synthetic users, where liked and disliked items replace the ground-truth interactions from the offline dataset. The results indicate minimal differences between the SimUSER-generated data and the real-world dataset. These differences can be attributed to real-world scenarios in which users are not constrained by the number of pages and the frequency of interactions with the recommender system. Interestingly, the results are consistent across different systems, with the ranking of models remaining the same between offline and SimUSER-generated metrics. Overall, these experimental results highlight that SimUSER can reliably measure traditional metrics while also enabling the exploration of system performance across different user demographic groups, various website settings (e.g., number of items per page), and different settings of the recommender system.

## F.10 Ablation Studies

### F.10.1 Impact of the Knowledge-Graph Memory on SimUSER

Here, our focus is on evaluating the impact of incorporating a knowledge-graph memory on the performance. Specifically, the goal is to determine whether employing the KG memory, which simulates external influences such as reviews, enhances believability in human proxies. All models follow the same settings as in Sec 4.3. Table 7, highlights that leveraging the KG structure significantly reduces both RMSE and MAE across different datasets. This module mirrors how our prior expectations of an item can shape and bias our assessment of it.

Table 7: Performance comparison in rating prediction for agents equipped with (top two rows ♥) and without a KG memory (bottom two rows ♣). Asterisks (*) denote statistically significant improvements when the KG memory is used.

| Methods | MovieLens | | AmazonBook | | Steam | |
|---------|-----------|-----|------------|-----|-------|-----|
| | RMSE | MAE | RMSE | MAE | RMSE | MAE |
| SimUSER(zero) ♥ | 0.6112* | 0.5353* | 0.6698 | 0.5597* | 0.6844* | 0.6392* |
| SimUSER(zero) ♣ | 0.6712 | 0.6400 | 0.6943 | 0.6220 | 0.7203 | 0.6772 |
| SimUSER(sim) ♥ | **0.5341*** | **0.4671*** | **0.5919*** | **0.4562*** | **0.6153*** | **0.5686*** |
| SimUSER(sim) ♣ | 0.6387 | 0.6551 | 0.6295 | 0.4996 | 0.6654 | 0.6481 |

Table 8: Performance of Persona Matching in Predicting Age and Occupation Using the MovieLens-1M Dataset.

| Metric | Age | Occupation |
|--------|-----|-----------|
| Accuracy | 0.7230 | 0.6764 |
| Precision | 0.7586 | 0.6933 |
| Recall | 0.7921 | 0.7430 |
| F1 Score | 0.7749 | 0.7172 |

### F.10.2 PERSONA MATCHING: AGE, OCCUPATION

In this study, we postulate that personas are crucial for capturing the heterogeneity and diversity present in real-world recommender networks. These attributes significantly shape individual behaviors and preferences, which subsequently influence the overall dynamics of the system. To evaluate the effectiveness of our self-consistent persona-matching technique, we conducted an experiment using the MovieLens-1M dataset. The goal was to predict the age and occupation of users based on their historical data. This task was formulated as a classification problem. Our results are summarized in table 8. We observe a high degree of alignment between the predicted and actual user personas, highlighting the effectiveness of Phase 1 in SimUSER. Overall, *persona matching* turns out to be reasonably robust for enriching simulated agents with detailed backgrounds, including domains where explicit demographic data is not readily provided.

### F.10.3 PERSONA MATCHING: PERSONALITY

In order to assess the quality of persona matching in predicting personality traits from historical interaction data, we conduct an additional experiment using the Personality 2018 dataset Nguyen et al. (2018). The primary objective is to evaluate whether our model could accurately infer users' Big Five personality traits — Openness, Conscientiousness, Extraversion, Agreeableness, and Neuroticism, based solely on users' watching history. For a fair comparison, the personality traits within the dataset, as well as the predictions, are normalized to a scale ranging from 0 to 1. We report the results for various lengths of movie history $\varrho \in \{10, 20, 50\}$. This task is formulated as a regression problem, with the model's performance evaluated using the Mean Absolute Error (MAE) for each personality trait prediction. Figure 11 summarizes the results, showing that our model achieved an average MAE of $0.155$ across all traits. Besides, the results reveal that using a history length of 50 items reduces the average MAE from $0.279$ (10 items) to $0.155$, demonstrating that self-consistent persona matching can reasonably predict personality traits of users from their past interactions.

### F.10.4 CHOICE OF FOUNDATION MODEL

We seek to evaluate the performance of our methodology using various foundation models on the movie rating task. Specifically, we compare the results obtained by employing GPT-4o-mini, GPT-4o, Mistral-7b Instruct, Llama-3 Instruct, and Phi-3-mini as the underlying LLMs. The results, presented in Table 9, demonstrate that the performance of SimUSER is generally robust across different foundation models. While GPT-4o exhibits significantly lower mean RMSE and MAE (t-test $p < 0.05$), GPT-4o-mini achieves similar performance but with a lower inference time. Mistral-

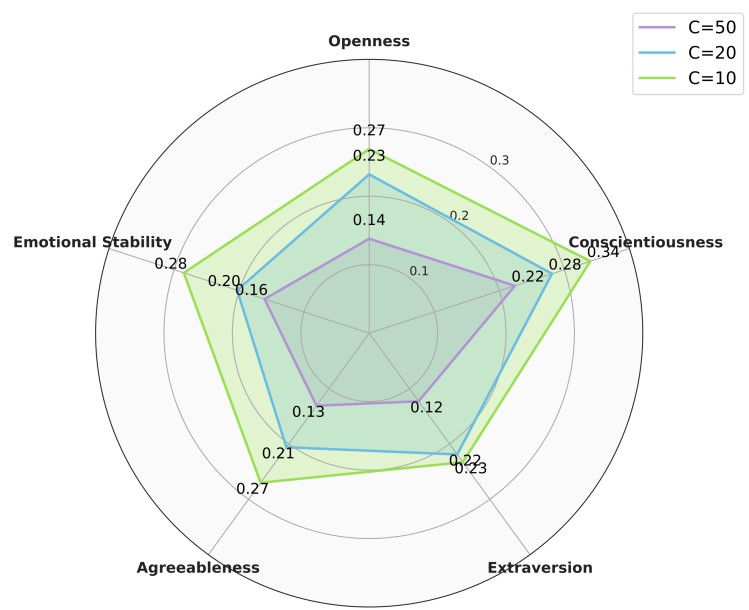

Figure 11: MAE of personality trait predictions for different C values.

Table 9: Performance comparison in rating prediction on MovieLens with different types of foundation LLMs.

| Methods | MovieLens | | AmazonBook | | Steam | |
|---|---|---|---|---|---|---|
| | RMSE | MAE | RMSE | MAE | RMSE | MAE |
| GPT-4o-mini | 0.5341 | 0.4671 | 0.5919 | 0.4562 | 0.6153 | 0.5686 |
| GPT-4o | **0.4866** | **0.4291** | **0.5656** | **0.4124** | **0.5679** | **0.4981** |
| Mistral-7b Instruct | 0.5623 | 0.5019 | 0.6588 | 0.5061 | 0.6552 | 0.6436 |
| Llama-3 Instruct | 0.6034 | 0.5953 | 0.6487 | 0.4869 | 0.6600 | 0.6482 |
| Phi-3-mini | 0.6486 | 0.6100 | 0.6925 | 0.5932 | 0.7317 | 0.7090 |

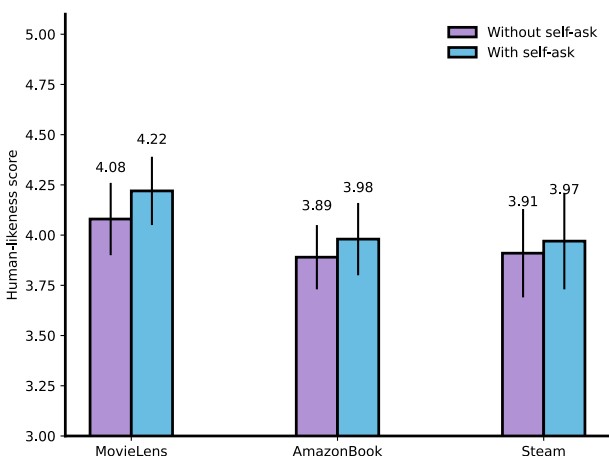

Figure 12: Human-likeness score evaluated by GPT-4o for SimUSER trained without and with self-ask document retrieval.

7b Instruct also performs reasonably well on the MovieLens dataset. On the other hand, Llama-3 Instruct and Phi-3-mini, while competitive, show higher errors.

### F.10.5 EPISODIC MEMORY ABLATION

In this study, we focus on the episodic memory. We postulate that the proposed self-ask strategy for retrieving documents contributes to the believability of synthetic users. We evaluate agents that use a plain memory, retrieving documents based on the embedding of the query, against agents employing the self-ask strategy. An LLM evaluator is employed to measure the believability of generated interactions. Across all domains, the agents employing self-ask retrieval are more realistic than their simple counterparts (Figure 12), emphasizing the advantages of our episodic memory architecture. This believability gain is largely attributed to the retrieval of more diverse yet relevant "context" during decision-making.

### F.10.6 IMPACT OF PERCEPTION MODULE

Finally, we investigate the impact of the perception module on the believability of SimUSER agents. From Table 10, we can draw several conclusions. Agents consistently exhibit more realistic behavior when employing a perception module (♠), likely due to the inclusion of visual details and unique selling points associated with the items. Another observation is that the believability gain is lower on AmazonBook than other datasets. This may be attributed to the nature of books as products — users are less likely to judge a book based on its cover and more inclined to make decisions based on its description.

By looking closely at the interactions, we also noticed that agents with behaviors of users featuring different personas are significantly driven by emotional tones provided in the captions. For instance, an agent with high openness may be more inclined to select movies with captions that use positive language like "exciting" or "inspiring". We also postulate that SimUSER (♦) may inherit biases from the LLM's interpretation of item descriptions, although these biases could be partially mitigated through adding factual information based on the captions. This suggests that the perception module not only enhances realism at a macro level but also contributes to more visually and emotionally driven engagement.

## G ADDITIONAL BASELINE INFORMATION

In this section, we present a comparative analysis of our method against two widely recognized baselines for simulating user interactions: RecAgent and Agent4Rec. Additionally, we examine

Table 10: Human-likeness score evaluated by GPT-4o for SimUSER without (♦) and with (♠) perception module. Asterisks (*) denote statistically significant improvements when the perception module is activated.

|  | **MovieLens** | **AmazonBook** | **Steam** |
|---|---|---|---|
| RecAgent | $3.01 \pm 0.14$ | $3.14 \pm 0.13$ | $2.96 \pm 0.17$ |
| Agent4Rec | $3.04 \pm 0.12$ | $3.21 \pm 0.14$ | $3.09 \pm 0.16$ |
| SimUSER (♦) | $4.12 \pm 0.15$ | $3.95 \pm 0.16$ | $3.88 \pm 0.19$ |
| SimUSER (♠) | $\mathbf{4.22 \pm 0.17}$* | $\mathbf{3.98 \pm 0.18}$* | $\mathbf{3.97 \pm 0.24}$* |

the features of RecMind-SI, which is capable of estimating item ratings. Table 11 offers a detailed comparison of these methods with human users.

Table 11: Comparison of our method with prior approaches. The 'Emotion/Mood' column represents the agent's capability to incorporate emotional and mood-based reasoning, while 'External Factors' reflects the model's ability to make decisions informed by its pre-existing assumptions about items.

| Name | Interactive | Memory | Image Perception | Emotion/Mood | External Factors | Affordable |
|---|---|---|---|---|---|---|
| RecMind-SI | ✗ | Personalized + World Knowledge | ✗ | ✗ | ✓ | ✓ |
| RecAgent | ✓ | Short Term + Long Term | ✗ | ✗ | ✗ | ✓ |
| Agent4Rec | ✓ | Episodic | ✗ | ✓ | ✗ | ✓ |
| SimUSER | ✓ | Knolwedge Graph + Episodic | ✓ | ✓ | ✓ | ✓ |
| Human | ✓ | Brain | ✓ | ✓ | ✓ | ✗ |

Note that, in our analysis, we consider human users as not *affordable* when compared to the significantly lower cost of leveraging synthetic agents.

## H  DISCUSSION

Our study verifies the possibility of synthetic users in recommendation. SimUSER has demonstrated its effectiveness to generate faithful users across a wide range of recommendation domains, including movies, books, and games. However, we also acknowledge that our method has certain limitations.

Observed behaviors are well-aligned with existing theories and common behaviors in the recommendation domain. Phenomena at micro-level (rating, watching) are manifestations of agent endogenous behaviors. But why agents possess these behaviors are unexplored due to the black-box nature of the large language models we adopted. A potential reason could be that LLMs are trained on a massive corpus that includes texts from various domains.

This paper introduces a strategy to capture external factors in recommendation via a knowledge graph memory. The key is for the system to retrieve similar items based on the structure of the graph and their semantic similarities. But in future work, we aim to explore the full potential for knowledge-graph enriched prompts, such as by dynamically breaking down a complex problem into smaller subproblems Besta et al. (2024) and retrieving relevant information from the graph using LLM-generated queries.

The episodic module relies on a self-ask strategy to retrieve documents. Although this technique introduces additional overhead, we found that it also enhances the diversity and quality of retrieved documents by extrapolating documents relevant to the query, which is of critical importance in recommendation systems, where granular user preferences depend on accessing a comprehensive range of relevant information.

LLMs may replicate biases prevalent in social spaces, such as some groups of individuals being underrepresented. This is problematic if it causes designers to then underlook these peoples' needs when designing a recommender system. In our experiments, we mitigated this risk by ensuring a broad range of personas via diverse potential occupations, age, and personalities. We also measured

Table 12: Average running time of Agent4Rec and SimUSER without (♥) and with(♣) parallelization of LLM calls.

| Methods | Average Running Time (h) |
| --- | --- |
| Agent4Rec(♥) | 9.3 |
| Agent4Rec(♣) | 0.51 |
| SimUSER(♥) | 9.8 |
| SimUSER(♣) | 0.58 |

the discrepancy between identified and real personas. Our future investigation will focus on analyzing underrepresented user groups, as well as evaluating persona matching on a wider range of domains (e.g., food).

In terms of evaluation, the assessment of synthetic users' behavior in this study was aimed at a range of tasks like rating items, measuring the alignment between their feelings and ratings, and LLM-based believability. While these tasks are relevant and provide insight into the model's performance, they may not fully capture the complexity of real-world user interactions. Real users engage with systems in multifaceted ways, often influenced by contextual factors, evolving preferences, and interactions that extend beyond simple task execution. To address this limitation, future work could involve direct comparisons with real user performance, particularly at a macro level. We also seek to include human-based evaluations to assess the believability of user interactions.

Finally, UX and UI drive our choices and actions in real-world applications. Our simulation, on the other hand, does not fully replicate all those intricate factors, which introduces a gap between real life and simulation. An important future direction is developing an image-based simulator to better capture the complex nature of user experience.

## I    COST ANALYSIS

We report here the cost of running SimUSER in terms of "cost per 1000 users". Given that the cost per agent may fluctuate slightly due to variations in the number of interactions and the outputs generated by the LLM, we focus on the total cost per 1000 users. Note that the cost per agent rises approximately linearly with the number of agents. Although we express this cost in USD, it may vary significantly depending on the choice of LLM. For our implementation, we utilized GPT-4o-mini from OpenAI as the underlying model.

As a result, SimUSER achieves a rather low cost of approximately $12 ($0.0012/User). A simulation involving 1,000 Agent4Rec agents would cost approximately $10 ($0.0010/User). The cost difference mainly stems from the integration of images to enable visual-driven reasoning. Overall, both methods have similar costs and can be employed as *affordable* proxies for human interaction, particularly when compared to their human counterparts. Even though our method incurs a slightly higher cost compared to Agent4Rec, we believe the enhanced believability and greater depth of interactions it offers are key advantages compared to Agent4Rec.

## J    RUNNING TIME ANALYSIS

In this section, we provide a running time comparison between the proposed method and Agent4Rec. The simulation involved 1,000 users interacting with the movie recommender system, using GPT-4o as the base model. We report the results both without (♥) and with (♣) parallelization of LLM calls. For the parallelized version, the maximum number of workers was set to 500. The results in Table 12 highlight that SimUSER and Rec4Agent achieve similar inference speeds. As expected, parallelizing the LLM calls significantly reduces the inference time for SimUSER (♣). This demonstrates two key points: 1) user interactions can be scaled to a larger number of users (1,000+) while maintaining a low simulation time, 2) the present approach can be parallelized ♣ to further reduce the running time.

## K  THUMBNAIL CAPTION EXAMPLE

We provide below an example of a caption generated by the perception module, given as input a poster of the movie "GoldenEye (1995)":

---

**Output Perception Module**

**Input:** Poster of the movie "GoldenEye (1995)"
**Caption:** The movie poster for "GoldenEye 007" prominently features James Bond, portrayed by Pierce Brosnan, in the center, aiming a gun directly at the viewer, conveying a sense of intense action and suspense. To the left, a close-up of a mysterious woman with a confident expression adds intrigue, hinting at her pivotal role in the plot. A helicopter and a fighter jet, both mid-flight, further emphasize the film's thrilling, fast-paced narrative. The iconic "007" logo, with the gun symbol integrated into the number, is prominently displayed at the bottom, reinforcing the film's connection to the legendary spy franchise. The overall mood of the poster is intense and action-packed, appealing to fans of the James Bond series and those looking for a thrilling cinematic experience.

---

## L  ETHICS AND REPRODUCIBILITY STATEMENT

This paper proposes an LLM-empowered agent framework designed to simulate user interactions with recommender systems in a realistic and cost-effective manner. While our approach offers significant benefits in terms of scalability and efficiency, it also raises ethical considerations. The use of such agents could lead to unintended consequences, such as bias amplification, where the synthetic agents might inadvertently reinforce existing stereotypes or present skewed recommendations due to biases in the training data. Additionally, there is a risk of manipulation of user preferences, as the synthetic agents could be used to subtly influence user behavior by consistently promoting certain types of content without explicit user consent. Furthermore, simulating interactions at a broad scale could result in the identification and exploitation of behavioral patterns that might encourage specific user behaviors, potentially leading to negative societal impacts. Finally, there is a concern that developers or designers might use synthetic users and displace the role of humans and system stakeholders in the design process.

Regarding privacy, SimUSER leverages LLMs' reasoning ability and factual knowledge without finetuning LLMs, ensuring that LLMs do not retain or remember user-specific data. In addition, employing matched personas instead of user-specific information available such as the occupation or gender prevents the leakage of user privacy information through external APIs.

Furthermore, since our experiments and analyses are conducted in English, we do not claim that our findings are universally applicable across all languages. However, the SimUSER framework may be adaptable to other languages with suitable modifications.

We acknowledge these risks and advocate for responsible deployment, including transparency, user awareness, and ongoing monitoring to mitigate potential harm. We also suggest that synthetic uses should not be a substitute for real human input in studies and design processes. Rather, these agents should be leveraged during the initial design phases to explore concepts, especially in situations where recruiting human participants is impractical or where testing certain theories with real people could be challenging or pose risks. By adhering to these principles, we can ensure that the deployment of synthetic users in the wild is ethical and socially responsible.

In the interest of reproducibility, we have evaluated SimUSER and baseline methods using publicly available datasets and codebases. The code and scripts necessary to reproduce SimUSER and experiments are available at https://github.com/SimUSER-paper/SimUSER.

