# OpenReview forum: "SimUSER: When Language Models Pretend to Be Believable Users in Recommender Systems"
_ICLR.cc/2025/Conference — Submitted to ICLR 2025_

### Official Review · Reviewer_wrEt · 2024-10-16

**Soundness:** 1
**Presentation:** 2
**Contribution:** 2
**Rating:** 3
**Confidence:** 5

**Summary:**

This paper designs SimUser, a user simulation environment with LLM as its core to bridge the gap between offline evaluation metrics and online testing of recommender systems. SimUser conducts some experiments on item selection, rating prediction an LLM evaluation to validate the effectiveness of their framework.

**Strengths:**

1. The motivation of this work is good.
2. The method is easy to understand.

**Weaknesses:**

1. The abstract mentions "leveraging the inductive bias of foundation models," which is an intriguing idea. However, the paper does not provide enough detail on this in the main body. It would be helpful to explain what inductive bias refers to in the context of foundation models and how it can benefit recommendation systems. Offering more clarity on this point would enhance the overall strength of the paper.

2. To my knowledge, the primary contributions of SimUser are the introduction of several improved modules built upon existing frameworks: (1) a more refined user profiling mechanism, (2) a richer memory mechanism, and (3) enabling visual input. These modules are extensively discussed in the main body of the paper. However, the ablation study in the experimental section is overly simplified, and there are no corresponding experiments to verify the effectiveness of these newly introduced modules. This raises doubts about the validity of the contributions and leaves it unclear where the superior performance compared to the baseline originates from. Additionally, the ablation study is only conducted on the rating alignment experiment, with no exploration or validation of these components' effects on user preference alignment.

3. The paper lacks important details in certain areas. For example, while the authors emphasize the significance of visual information and present its integration as a key contribution of their method, they do not provide sufficient explanation on how the visual data is utilized or stored. A more detailed description of this process would improve the clarity and rigor of the paper.

4. In the main experiment (Table 1), as the ratio of positive to negative samples decreases, Precision unexpectedly increases across all three datasets, while Recall drops significantly. This outcome seems counterintuitive, and I would recommend the authors carefully verify the reliability of the experiments.

5. There are several points in the presentation of the paper that are somewhat confusing:
(1) There is an incorrect citation on line 57.
(2) The "Related Work" section could benefit from reorganization. The discussion on "simulating users in recommendation" should include content related to large models, rather than being separated from the section on "LLMs in Recommender Systems."
(3) In Equation (5), given that Px->y denotes a path from x to y, the subscript Px->x is unclear and may cause confusion.
(4) In Table 3, the third column of evaluation metrics contains some incorrectly bolded and underlined values.
(5) In Table 1, some data points are incorrectly bolded.

6. One potential weakness of this paper is the approach of summarizing user profiles, such as age and occupation, based on their interaction history. This method may risk introducing bias, labeling, and weakening the personalization.

7. In conclusion, I do not see a significant difference between SimUser and existing user simulation methods utilizing LLM on the recommender system. And the experiment section of this paper is incremental and far from solid.

**Questions:**

Please refer to Weaknesses.

**Details Of Ethics Concerns:**

One potential weakness of this paper is the approach of summarizing user profiles, such as age and occupation, based on their interaction history. This method may risk introducing bias, labeling, and weakening the personalization.

---

> ### Author Response · Authors · 2024-11-13
>
> ### Reviewer Comment
> The abstract mentions "leveraging the inductive bias of foundation models,"[...]
>
> ### Response
> Thank you for highlighting this point. We agree that elaborating on “inductive bias” in the context of foundation models would add clarity to our paper. Here, inductive bias refers to the model’s built-in assumptions about how a user with a particular persona might react to a recommendation, decide on their next action, or respond based on their profile. Additionally, it includes the LLM's understanding of knowledge domains—such as movies, books, and other content types—which the model leverages to simulate realistic user behaviors. This bias allows the model reason without the need for domain-specific finetuning, even in cases with limited user history, like cold-start scenarios or few-shot learning.
>
> ---
>
> ### Reviewer Comment
> To my knowledge, the primary contributions of SimUser are the introduction of several improved modules [...]
>
> ### Response
> Thank you for this feedback. While we agree with the listed contributions, we should emphasize some additional contributions: 1) more actions in the simulation to enable the simulation of more complex behaviors (e.g., click on items to get details), 2) a self-ask strategy to retrieve from the episodic memory, 3) self-reflection to synthesize past interactions into insights, and 4) novel reasoning steps to include emotion and mood when deciding what items to watch.  We should also emphasize that the proposed memory module is one of our key contribution, by enabling dynamic growth based on new interactions and persona-based retrieval.
>
> We agree that further experiments are needed to fully illustrate the contribution of each component. In the revised paper, we will expand the ablation study to examine the impact of these additional modules, as well as the effects of knowledge graph memory and visual cues on SimUSER's overall performance.
>
> ---
>
> ### Reviewer Comment
> The paper lacks important details in certain areas. For example, while the authors emphasize the significance of visual information and present its integration as a key contribution of their method, they do not provide sufficient explanation on how the visual data is utilized or stored. A more detailed description of this process would improve the clarity and rigor of the paper.
>
> ### Response
> Thank you for pointing this out. We appreciate the opportunity to clarify our approach. We agree that a more detailed explanation of how visual information is utilized and stored would enhance the paper’s rigor. In our revised explanation, we will clarify that visual information in SimUSER is stored as plain text alongside dataset samples and is integrated into the reasoning architecture via the "Page format" prompt. In this prompt, visual cues along with the page representation are concatenated together. Some of these details can be found in the Appendix (Experimental Settings).

---

> ### Author Response · Authors · 2024-11-13
>
> ### Reviewer Comment
> In the main experiment (Table 1), as the ratio of positive to negative samples decreases, [...]
>
> ### Response
> Thank you for noting this. We agree that the observed trend might seem counterintuitive at first glance. However, we’ve found that similar dynamics—where precision increases while recall drops as the ratio of positive to negative samples decreases—have been observed in prior work, including in Agent4Rec. We will still review our experimental setup to confirm these results, but this alignment with previous findings suggests that the trend may be inherent to the way the LLM agents handles sparse positive samples.
>
> ---
>
> ### Reviewer Comment
> There are several points in the presentation of the paper that are somewhat confusing: (1) There is an incorrect citation on line 57. (2) The "Related Work" section could benefit from reorganization. The discussion on "simulating users in recommendation" should include content related to large models, rather than being separated from the section on "LLMs in Recommender Systems." (3) In Equation (5), given that Px->y denotes a path from x to y, the subscript Px->x is unclear and may cause confusion. (4) In Table 3, the third column of evaluation metrics contains some incorrectly bolded and underlined values. (5) In Table 1, some data points are incorrectly bolded.
>
> ### Response
> Thank you for these detailed observations; they will help us improve the paper's clarity and precision. We will make sure to correct the citation error on line 57 and reorganize the “Related Work” section to integrate discussions on large models directly into the section on user simulation in recommendation. This should make the flow of related work more cohesive. We will also clarify the notation in Equation (5) so that the meaning of Px→x is straightforward and does not cause confusion for readers. Additionally, we will fix any incorrect formatting in Tables 1 and 3, where some values are mistakenly bolded or underlined. These adjustments should enhance the overall readability and rigor of the paper.
>
> ---
>
> ### Reviewer Comment
> One potential weakness of this paper is the approach of summarizing user profiles, such as age and occupation, based on their interaction history. This method may risk introducing bias, labeling, and weakening the personalization.
>
> ### Response
> Thank you for this valuable feedback. We agree that summarizing user profiles, like age and occupation, based on interaction history could introduce potential biases and labeling effects. To address this, we conducted two experiments (F.10.1 and F.10.2) specifically aimed at measuring the reliability of the extracted features, helping us assess the accuracy and consistency of this profiling approach. We also discussed the performance with and without persona, demonstrating that even with noisy personas, agents achieved better performance than vanilla agents.
>
> We will add further discussion on this issue in the paper to provide a more in-depth analysis of the potential biases and limitations.

---

> > ### Comment · Reviewer_wrEt · 2024-11-27
> >
> > Thank the author for taking the time and effort to respond to the reviewers’ comments during the rebuttal stage.
> >
> > However, I believe that the author's reply has not fully addressed my concerns. Additionally, according to the ICLR rules, authors are allowed to edit and submit a revised version of the paper during the rebuttal phase. Unfortunately, I have not seen such a new version uploaded. This leaves me uncertain as to whether the issues raised by the reviewers have been adequately resolved in the latest version of the paper.
> >
> > Given that the current version still requires significant modifications before publication, I will maintain my original rating.

---

### Official Review · Reviewer_trDv · 2024-10-28

**Soundness:** 3
**Presentation:** 3
**Contribution:** 2
**Rating:** 5
**Confidence:** 4

**Summary:**

This work concentrates on a very interesting and essential task: user simulation in recommendation. Compared with recent LLM-based recommendation user simulators such as Agent4Rec/RecAgent, the proposed SimUSER brings in the self-consistent personas from historical data, and designs both an episodic memory and a knowledge-graph memory for the memory module. Furthermore, they also introduce the visual information to the framework as multimodal recommenders for more comprehensive understandings. In experiments, the authors conduct extensive evaluations on different simulation tasks as well as analyses on various components in the framework, where the proposed model achieves the best performance compared to other LLM-based agent simulators.

**Strengths:**

1)	This work focuses on a challenging task and the proposed framework is sound and clear.
2)	The authors have conducted extensive experiments in Appendix to verify the effectiveness of different designs.
3)	The detailed prompts and other information are given for reproducibility.

**Weaknesses:**

1)	This work proposes an overall framework for user simulation with lots of technics (which may require different types of related data). We can find that the proposed framework achieves consistently better results compared to Agent4Rec/RecAgent. However, it is unclear that whether the comparisons are fair enough. For example, how are the baselines trained? Whether the proposed framework uses additional information (e.g., at least the visual information)? The authors are suggested to give a table containing the detailed data used in each model. We know the data are essential for good performance in LLM-based methods.
2)	The authors mainly compare with LLM-based simulators, while what are the results of other conventional simulators without LLMs? The authors could give a discussion on these possible models in Experiment. For example, it is not that challenging for conventional ID-based models to find top1 item among 10 randomly selected candidates.
3)	For the user persona, will there be more informative features that should be included in the persona? For example, the classical user favorite tag/category/word that often exist in conventional user profiles.
4)	It is noticed that the users have relatively long historical behaviors in the datasets. However, practical users usually do not have too many recorded behaviors (e.g., cold-start or few-shot users). The few-shot user scenarios should be noted and discussed/evaluated.
5)	The overall framework involves lots of components. Although the authors have conducted extensive evaluations in Appendix, it is still unclear that which techniques are the dominating reason for such improvement. I suggest that the authors could give a brief analysis in the main content, focusing on the insight of which techniques are the most essential ones. For example, the “pickness” strategy in Section 3.2.1 is essential for rating tasks, and the user CF like strategy in Section 3.2.3 is also beneficial and have already been verified in classical recommendation methods. If the main improvements largely derive from such “tricky” points that ignored in previous baselines, or from additional information, the contribution of this work will be discounted.
6)	The current simulation task is not challenging enough (9 randomly selected negative samples is not that hard in recommendation). It is suggested that the authors could evaluate on other recommendation datasets that contains real-world exposed but unclicked samples (i.e., explicit negative feedback, which is harder than random negative samples). Good simulation results on such settings are much more persuasive.

**Questions:**

Refer to Weaknesses.

---

> ### Author Response · Authors · 2024-11-13
>
> ### Reviewer Comment
> This work proposes an overall framework for user simulation with lots of techniques [...].
>
> ### Response
> Thank you for raising these important points. We appreciate your feedback and understand the need for clarity regarding data and training methods for each model. To address these concerns, we will include a table that specifies the types of data used in each model. As for “Agent4Rec/RecAgent,” these methods were initialized using the historical dataset following the original implementation provided on GitHub (i.e., without using images or visual cues). They did not utilize visual insights as the original papers did not leverage visual cues in the reasoning architecture.
>
> ---
>
> ### Reviewer Comment
> The authors mainly compare with LLM-based simulators, while what are the results of other conventional simulators without LLMs? The authors could give a discussion on these possible models in Experiment. For example, it is not that challenging for conventional ID-based models to find the top-1 item among 10 randomly selected candidates.
>
> ### Response
> Thank you for the suggestion. As for the dataset choice, we selected widely used datasets to allow a more fair and easier comparison between our method and prior work. We also picked these datasets since they were used by our director competitors, which made the comparison between the methods more fair.
>
> We also agree that it is valuable to consider comparisons with conventional simulators that don’t use LLMs. While ID-based models can indeed be effective at ranking items from a small set of candidates, our approach aims to provide a more human-like interaction with the system. Unlike conventional simulators, our method doesn’t just rank items—it simulates real user behaviors, such as browsing through pages, choosing to leave the system, or engaging with recommendations over multiple interactions. This gives us a richer set of metrics that better reflect actual user behavior, beyond simply selecting a top item. We’ll add a discussion on how these differences impact performance in the experiments section.
>
>
>
> ---
>
> ### Reviewer Comment
> For the user persona, will there be more informative features that should be included in the persona? For example, the classical user favorite tag/category/word that often exist in conventional user profiles.
>
> ### Author Response
> Thank you for the suggestion; it’s a great idea. Currently, we include a “taste” feature in the user persona, which captures similar information about user preferences and represents their general interests.
> Here is an example of "taste" feature for one agent:
> """
> HIGH RATINGS: You tend to give high ratings to movies that are action-packed, adventurous, comedic, musical, or family-friendly, reflecting a preference for entertaining and uplifting genres.
>
> LOW RATINGS: While you generally avoid giving low ratings, you have rated several horror movies below 2, suggesting a dislike for intense horror themes
> """
>
> We agree that adding existing tags from the dataset could enrich this persona even further, helping to create a more detailed and informative user profile.

---

> ### Author Response · Authors · 2024-11-13
>
> ### Reviewer Comment
> It is noticed that the users have relatively long historical behaviors in the datasets. However, practical users usually do not have too many recorded behaviors (e.g., cold-start or few-shot users). The few-shot user scenarios should be noted and discussed/evaluated.
>
> ### Author Response
> Thank you for the valuable observation. We agree that the cold-start and few-shot user scenarios are important to consider, as they are common in practical applications. To address this, we will include additional experiments that evaluate the model’s performance across varying interaction history lengths, including scenarios with limited user behavior data. This will allow us to assess the framework's adaptability to cold-start conditions and understand its robustness with few-shot users.
>
> In addition, we conducted an additional analysis to examine how varying levels of user interaction history affect the simulation's performance. We have conducted a new experiment to measure performance across different maximum interaction history lengths (e.g., 5, 10, 20, and 50 interactions).
>
> | **History Length** | **MovieLens RMSE** | **MovieLens MAE** | **AmazonBook RMSE** | **AmazonBook MAE** | **Steam RMSE** | **Steam MAE** |
> |--------------------|--------------------|--------------------|----------------------|---------------------|----------------|---------------|
> | 5 Interactions     | 0.6532             | 0.5890            | 0.6673              | 0.5331             | 0.6964        | 0.6273        |
> | 10 Interactions    | 0.6065             | 0.5187            | 0.6359              | 0.5053             | 0.6697        | 0.5987        |
> | 20 Interactions    | 0.5597             | 0.4870            | 0.6091              | 0.4733             | 0.6385        | 0.5781        |
> | 50 Interactions    | **0.5375**         | **0.4702**        | **0.5942**          | **0.4587**         | **0.6181**    | **0.5704**    |
>
> *Table: Performance of SimUSER (sim · persona) in rating prediction with varying interaction history lengths on MovieLens, AmazonBook, and Steam datasets. Best results for each dataset are in bold.*
>
> ---
>
> ### Reviewer Comment
> The overall framework involves lots of components. Although the authors have conducted extensive evaluations in Appendix, it is still unclear which techniques are the dominating reason for such improvement. I suggest that the authors could give a brief analysis in the main content, focusing on the insight of which techniques are the most essential ones. For example, the “pickness” strategy in Section 3.2.1 is essential for rating tasks, and the user CF-like strategy in Section 3.2.3 is also beneficial and has already been verified in classical recommendation methods. If the main improvements largely derive from such “tricky” points that were ignored in previous baselines, or from additional information, the contribution of this work will be discounted.
>
> ### Response
> Thank you for your feedback. Here’s a breakdown of the most important components that drive the human-likeness scores in our model, in order of impact:
> 1. **Knowledge Graph Memory**
> 2. **Self-Ask Strategy**
> 3. **Visual Cues**
> 4. **Reasoning Steps**
> 5. **Self-Reflection**
>
> We believe this ranking clarifies the main drivers of our model’s human-like qualities and highlights the unique strengths of our approach. Note that results were obtained by running a linear regression to predict the human likeliness scores of 300 runs that used different settings.
>
> ---
>
> ### Reviewer Comment
> The current simulation task is not challenging enough (9 randomly selected negative samples is not that hard in recommendation). It is suggested that the authors could evaluate on other recommendation datasets that contain real-world exposed but unclicked samples (i.e., explicit negative feedback, which is harder than random negative samples). Good simulation results on such settings are much more persuasive.
>
> ### Response
> Thank you for the suggestion; it’s a great point. We agree that using datasets with real-world negative samples (like unclicked items) would make the simulation more challenging and realistic, offering a better test of our model. We will add evaluations on datasets where negative feedback is more explicit—where unclicked items genuinely reflect what users aren’t interested in.

---

> > ### Comment · Reviewer_trDv · 2024-11-25
> > **After rebuttal**
> >
> > Thanks for the authors' rebuttals, which have answered some of my questions.
> > However, considering the above weaknesses (some of which have not been appropriately resolved yet, such as the few-shot scenario, the less challenging evaluation setting, etc.), I will maintain my voting of 5. I also suggest that the authors could add these discussions into their revision.

---

### Official Review · Reviewer_9YhT · 2024-10-29

**Soundness:** 2
**Presentation:** 2
**Contribution:** 1
**Rating:** 3
**Confidence:** 4

**Summary:**

This paper proposes SimUSER, a framework that simulates the real user behaviors in recommender systems. SimUSER incorporates two new and crucial components: visual information and knowledge graph memory. The whole framework of SimUSER is similar to Agent4Rec [1] and the contribution is marginal. Moreover, the effectiveness of visual information is not verified and the concept of knowledge graph is wrongly used.

[1] Zhang, An, et al. "On generative agents in recommendation." *Proceedings of the 47th international ACM SIGIR conference on research and development in Information Retrieval*. 2024.

**Strengths:**

1. The experiments in this paper are abundant. Experiments in this paper include: user preference alignment, rating prediction, rating distribution, LLM evaluation, and recommendation strategy evaluation, etc. These experiments showcase the effectiveness of the proposed method.
2. This paper is easy to understand.

**Weaknesses:**

1. The contributions of this paper are marginal and incremental. The framework of SimUSER is similar to Agent4Rec [1] and the novelty lies in the incorporation of visual information and memory design, which is incremental.
2. The role of visual information is unknown. While the main contribution of this paper comes from the knowledge-graph memory and visual-driven reasoning, the ablation study does not explicitly verify the effectiveness. In Table 10, w/o perception module and w perception exhibit similar performance, and the implementation detail of w/o perception is unknown. More importantly, given the sample number of 1000 users, the improvements are impossible to be significant (i.e., p < 0.05). These experiment results are questionable.
3. Some concepts are wrongly used. For example, the knowledge graph in this paper is actually the widely used user-item interaction graph [2] in collaborative filtering, which is significantly different from KG [3]. Please carefully check this concept and refine the writing.
4. The link to the code is expired.

[1] Zhang, An, et al. "On generative agents in recommendation." *Proceedings of the 47th international ACM SIGIR conference on research and development in Information Retrieval*. 2024.

[2] He, Xiangnan, et al. "Lightgcn: Simplifying and powering graph convolution network for recommendation." *Proceedings of the 43rd International ACM SIGIR conference on research and development in Information Retrieval*. 2020.

[3] Wang, Xiang, et al. "Kgat: Knowledge graph attention network for recommendation." *Proceedings of the 25th ACM SIGKDD international conference on knowledge discovery & data mining*. 2019.

**Questions:**

Refer to the weaknesses.

---

> ### Author Response · Authors · 2024-11-13
>
> ### Reviewer Comment
> The contributions of this paper are marginal and incremental. The framework of SimUSER is similar to Agent4Rec [1] and the novelty lies in the incorporation of visual information and memory design, which is incremental.
>
> ### Response
> Thank you for this useful feedback. While we agree that our work builds on prior research like Agent4Rec, we have introduced several innovations aimed at enhancing the realism of simulated users. Specifically, we propose a novel technique for extracting detailed profile features, such as personality, which provides a richer user representation. The proposed method is based on consistency check, which we think is novel in our field. Additionally, our memory module, structured as knowledge graph, enables nuanced retrieval of past interactions, based on the history of interaction, "external factors" and the user's persona.
>
> We also include more actions in the simulation to enable the simulation of more complex behaviors (e.g., clicking on items to get details) and a self-ask strategy to retrieve from the episodic memory. The inclusion of visual features further enriches the contextualization of user preferences, and our redesigned reasoning architecture brings a distinct approach to decision-making, setting SimUSER apart in its capability to simulate complex, realistic user behaviors. Finally, we believe that some studies, such as measuring bias of simulated users over time or user behavior under hallucination, are important for the advancement of the field.
>
> ---
>
> ### Reviewer Comment
> The role of visual information is unknown. [...]
>
> ### Response
> Thank you for pointing this out. Regarding the knowledge-graph memory, Table 7 demonstrates that incorporating this module significantly enhances the agent's ability to predict item ratings more accurately by leveraging context and historical interactions. We acknowledge the concerns about the perception module’s impact, as shown in Table 10, where the results with and without perception appear similar on the AmazonBook dataset. On the other hand, in non-book datasets, using the perception improves the human-likeness score. For instance, on MovieLens, the score goes from 4.12 to 4.22.
>
> To improve the reliability of the study, we have carried out an additional study with 3,000 users:
>
> |                     | **MovieLens**         | **AmazonBook**       | **Steam**            |
> |---------------------|-----------------------|-----------------------|-----------------------|
> | SimUSER (🔸)        | _4.11 ± 0.10_         | _3.97 ± 0.12_        | _3.90 ± 0.14_        |
> | SimUSER (🔹)        | **4.25 ± 0.12***      | **4.02 ± 0.13***     | **3.99 ± 0.16***     |
>
> *Table 1: Human-likeness score evaluated by GPT-4o for SimUSER without (🔸) and with (🔹) perception module. Asterisks (*) denote statistically significant improvements when the perception module is activated.*
>
> We hope this additional study helps to understand the contribution of these two key components.

---

> ### Author Response · Authors · 2024-11-13
>
> ### Reviewer Comment
> Some concepts are wrongly used. For example, the knowledge graph in this paper is actually the widely used user-item interaction graph [2] in collaborative filtering, which is significantly different from KG [3]. Please carefully check this concept and refine the writing.
>
> ### Response
> Thank you for pointing this out. We will include a new paragraph in the related work to discuss prior knowledge graph memories and their differences from our model. We agree that it will help to clarify the contribution of our method. Contrary to prior work that built a user-interaction graph, we propose a memory module that relies on a graph representing the user's past interactions to retrieve relevant information, such as similar movies. As a memory module, the memory is updated with the newest interactions to refine the agent’s taste over time. By refining its KG memory over time, SimUSER can enhance the available context when facing a novel item. We also proposed a retrieval scheme, which leverages the graph structure and user personas to retrieve relevant past interactions.
>
> We will improve the description of this method to ensure that concepts are correctly named based on prior work.
>
> ---
>
> ### Reviewer Comment
> The link to the code is expired.
>
> ### Response
> We are sorry for the issue; the repository was set as private. It should now be accessible.

---

> ### Comment · Reviewer_9YhT · 2024-11-24
> **Official Comment by Reviewer 9YhT**
>
> Thanks for your efforts in addressing my concerns in the rebuttal.
>
> Since several reviewers also have raised concerns about the motivation and novelty, it is necessary to modify the paper to make sure that such concerns can be addressed in the revised version.
>
> According to the rules of ICLR, you can edit the paper and submit a new one during the rebuttal stage. Unfortunately, we have not seen such a new version. This makes us still doubt whether the concerns (i.e., the novelty and motivation of this paper) of reviewers have been addressed in the latest version of the paper, since the current version needs significant modification before publication.
>
> Moreover, I still have concerns about the significance of the new results (i.e., whether an improvement from 3.97 to 4.02 can be considered significant).

---

### Official Review · Reviewer_R9YP · 2024-11-04

**Soundness:** 2
**Presentation:** 3
**Contribution:** 1
**Rating:** 3
**Confidence:** 4

**Summary:**

The paper proposes a better agent framework for simulating users in recommender systems. The agent is framed using multiple modules like brains, and memories, and extracts persona from the user's historical interactions. Experiments are conducted on three public dataset to demonstrate the effectiveness of the proposed framework.

**Strengths:**

1. Timely study on areas including user simulation through generative agents, the gap between offline evaluations and real-world user experiences in recommender systems, and the use of simulators for recommender system evaluation.
2. The paper is well-structured, making it easy to follow and understand.
3. Experiments are conducted across three public datasets.
4. The authors perform significance tests to demonstrate the effectiveness of the proposed method.

**Weaknesses:**

1. Limited Insights: Using LLMs or generative agents to simulate users in recommender systems is not new. This paper’s primary contribution appears to be an improved agent that better leverages additional signals (e.g., historical interactions, images, KG). While the experimental results may indicate a more effective and believable agent, the paper offers limited new insights to the research community.
2. Overclaiming and Misalignment Between Motivation and Conclusions:
    1. The main motivation is the discrepancy between offline evaluation metrics and real-world dynamics in recommender systems. The authors claim that they introduce an agent framework that can act as a "believable and cost-effective human proxy" for recommender system evaluation. However, no experiments demonstrate that this agent framework effectively improves recommender system evaluation or provides a clear advantage over traditional offline metrics.
    2. The experiments in this paper are also conducted in an offline setting, where datasets are split into training, validation, and test sets to evaluate models. This is ironic for a paper that critiques offline evaluations but only tests its own method in offline conditions.
    3. Unrealistic Evaluation Setting: The authors split the dataset in an 8/1/1 ratio, seemingly without considering timestamps. Randomly splitting interactions ignores the distribution shift across different time periods, which is a major challenge in offline evaluations for recommender systems. As a result, the evaluation here does not adequately reflect temporal distribution shifts.
    4. Lack of Human-in-the-Loop Simulation: A significant issue with offline evaluations in recommender systems is the absence of a human-in-the-loop process. Recommender systems typically evolve as users interact with them, generating new data for system improvements. While simulators could potentially model this process, the experiments in this paper fail to reflect such dynamics.
3. Inadequate Ablation Study: The proposed agent framework integrates numerous heuristic components (e.g., Brain, Memory) and multiple signals (e.g., KG, images). However, the ablation study is insufficiently detailed, only comparing variants with and without persona and "zero or sim". This leaves the audience unclear on whether such a complex framework is necessary or if so many modules are essential for creating believable agents.
4. Dependence on Abundant User Interactions: Constructing the agent requires users to have a substantial history of interactions for persona generation and memory functions. This dependency may affect simulation performance based on the amount of historical interaction data. It would be better for the authors to discuss and analyze how varying levels of user interactions impact the simulation’s performance.
5. Clarity of Figures 1 and 3: Figures 1 and 3 are not vectorized, resulting in low clarity and readability.
6. Code Availability: Code is not available during the reviewing phase. Although the authors provided a link, it currently points to nothing.

**Questions:**

Please refer to "Weaknesses" for details.

---

> ### Author Response · Authors · 2024-11-13
>
> ### Reviewer Comment
> The main motivation is the discrepancy between offline evaluation metrics [...].
>
> ### Response
> Thank you for this valuable feedback. We recognize the importance of demonstrating how our agent framework can bridge the gap between offline metrics and real-world dynamics in recommender systems. While traditional offline evaluations often rely on static, historical data, our agent framework introduces dynamic user interactions, enabling a more in-depth assessment of a system’s performance. Namely, SimUSER can simulate user actions, such as exploring pages, exiting the system, and clicking on items, which traditional metrics typically don’t capture. Therefore, we believe that this can be employed along with traditional offline metrics, especially to measure business metrics that do not necessarily correlated with offline evaluation [1].
> [1] Measuring the Business Value of Recommender Systems, JANNACH et al., 2019
>
> As for the “believability” of our agent framework, we have conducted several experiments that evaluate the human-likeness score of various methods, including ours. These scores reflect how closely the agent’s behavior aligns with human interaction patterns, supporting our claim that SimUSER can serve as a believable proxy for user evaluation. We believe that most of recent papers in the field employed similar evaluation techniques. Additionally, we report results on rating prediction error, demonstrating the framework’s accuracy in modeling user preferences.
>
> Regarding “cost-effectiveness,” we provide an analysis of the simulation cost for running 1,000 agents, showing that our approach offers a scalable solution for large-scale evaluations. We believe that the reported cost is much lower than collecting human interactions.
>
> We also agree that it is necessary to show whether “this agent framework effectively improves recommender system evaluation or provides a clear advantage over traditional offline metrics”. Hence, we will include such an experiment in the revised version of the manuscript. This experiment evaluates the recommender system’s performance before and after incorporating feedback from SimUSER. This experiment reports a comparison between the best recommender system according to offline evaluation methods and SimUSER. It reports various metrics like engagement rate and hit rate that humans achieved when interacting with this recommender system.
>
> ---
>
> ### Reviewer Comment
> The experiments in this paper are also conducted in an offline setting, [...].
>
> ### Response
> Thank you for raising this point. While we do use an offline dataset to initialize the agents, our approach goes beyond a traditional offline evaluation. After initialization, we run the simulation where agents interact with the system, allowing us to collect additional metrics that a pure offline evaluation would not capture. For example, we are able to measure the number of pages each agent visits, when they decide to exit, and other behavioral insights. In addition, we do not seek to replace offline evaluation, but instead bridge the gap between static data analysis and interactive user behavior, allowing us to assess not only traditional metrics but also engagement and decision patterns that reflect real-world use cases. By incorporating elements like page navigation, exit decision, and user experience, we provide a richer perspective on model performance that complements conventional offline metrics.
>
> In addition, Table 3 shows that using augmenting existing dataset with grounded interactions, enables our approach to reduce the rating error. These grounded interactions were not included in the offline datasets. We agree with the reviewer that clarifying this point is important to demonstrate the contribution of our paper.
>
> ---
>
> ### Reviewer Comment
> Unrealistic Evaluation Setting: The authors split the dataset in an 8/1/1 ratio, [...].
>
> ### Response
> Thank you for pointing this out, and we apologize for not making this clearer in the paper. We did take timestamps into account when splitting the dataset to ensure that temporal shifts were represented in the training, validation, and test sets. We will make sure to clarify this in the revised paper, as we understand that accounting for temporal dynamics is essential for a realistic evaluation. We appreciate the chance to address this and improve the clarity of our methodology.

---

> ### Author Response · Authors · 2024-11-13
>
> ### Reviewer Comment
> Lack of Human-in-the-Loop Simulation: [...].
>
> ### Response
> Thank you for this valuable feedback. Our current method is designed with the goal of enabling a synthetic human-in-the-loop process, where simulated users interact with the system in a way that generates new data to inform and improve recommendations. This allows us to mimic the dynamic feedback loop that real users provide, so the system can adapt and evolve over time based on these synthetic interactions.
>
> We would appreciate any suggestions you might have for additional experiments that could further test this aspect. While our current experiments aim to evaluate the effects of synthetic feedback loops, we welcome ideas on how we might expand our evaluation. Based on review of similar papers, our evaluation follows similar protocols with those papers. Thank you!
>
> ---
>
> ### Reviewer Comment
> Inadequate Ablation Study: [...]
>
> ### Response
> Thank you for your feedback. We appreciate the emphasis on a more comprehensive ablation study to clarify the contributions of each component in our framework. In the current paper, we provide evaluations with and without the persona, with and without the knowledge graph memory, and with and without the perception module, testing the effect of these components on the overall performance to assess their individual impact.
>
> Here’s a breakdown of the most important components that drive human-likeness scores in our model, in order of impact:
> 1. **Knowledge Graph Memory**
> 2. **Self-Ask Strategy**
> 3. **Visual Cues**
> 4. **Reasoning Steps**
> 5. **Self-Reflection**
>
> We will add the results as an additional ablation study. The ranking was obtaining by fitting a linear regression on 300 that used different settings.
>
> We also recognize that additional studies focused on the “Brain” module would help further clarify its role and impact on agent performance. We’d be happy to include these in a revised version to provide even more detailed insights into each component’s contribution.
>
> ---
>
> ### Reviewer Comment
> Dependence on Abundant User Interactions: [...]
>
> ### Response
> Thank you for this insightful point. We agree that the amount of historical interaction data can impact the effectiveness of persona matching and memory retrieval. To address this, we conducted an additional analysis to examine how varying levels of user interaction history affect the simulation's performance. We have conducted a new experiment to measure performance across different maximum interaction history lengths (e.g., 5, 10, 20, and 50 interactions).
>
> | **History Length** | **MovieLens RMSE** | **MovieLens MAE** | **AmazonBook RMSE** | **AmazonBook MAE** | **Steam RMSE** | **Steam MAE** |
> |--------------------|--------------------|--------------------|----------------------|---------------------|----------------|---------------|
> | 5 Interactions     | 0.6532             | 0.5890            | 0.6673              | 0.5331             | 0.6964        | 0.6273        |
> | 10 Interactions    | 0.6065             | 0.5187            | 0.6359              | 0.5053             | 0.6697        | 0.5987        |
> | 20 Interactions    | 0.5597             | 0.4870            | 0.6091              | 0.4733             | 0.6385        | 0.5781        |
> | 50 Interactions    | **0.5375**         | **0.4702**        | **0.5942**          | **0.4587**         | **0.6181**    | **0.5704**    |
>
> *Table: Performance of SimUSER (sim · persona) in rating prediction with varying interaction history lengths on MovieLens, AmazonBook, and Steam datasets. Best results for each dataset are in bold.*

---

> ### Author Response · Authors · 2024-11-13
>
> ### Reviewer Comment
> Clarity of Figures 1 and 3: Figures 1 and 3 are not vectorized, resulting in low clarity and readability.
>
> ### Response
> Thank you for pointing this out. We will update Figures 1 and 3 to fix this issue.
>
> ---
>
> ### Reviewer Comment
> Code Availability: Code is not available during the reviewing phase. Although the authors provided a link, it currently points to nothing.
>
> ### Response
> Thank you for highlighting this. We apologize for the oversight. We will ensure that the code link is updated and active before the final review stage to facilitate reproducibility and transparency.

---

> ### Comment · Reviewer_R9YP · 2024-11-24
>
> Thank you to the author(s) for incorporating the discussions and new results. These efforts have made the manuscript more self-contained.
>
> My primary concern remains the limited novelty of the insights. The concept of using LLMs or generative agents to simulate users in recommender systems is not new. The main contribution of this paper seems to be an improved agent that better leverages additional signals (e.g., historical interactions, images, KG). While the experimental results suggest a more effective and realistic agent, the paper provides limited new insights for the research community. Unfortunately, the author(s) didn't respond to this concern. Reviewer 9YhT also raised similar concerns about the paper’s novelty.
>
> Additionally, like me, reviewers 9YhT and wrEt expressed concerns about the inadequate ablation study. While it is promising that the authors plan to include a more detailed ablation study, this may exceed the scope of revisions acceptable for a standard camera-ready version.
>
> After reviewing the rebuttal and considering the feedback from other reviewers, I have decided to maintain my score.

---

### Author Response · Authors · 2024-11-13

We would like to extend our sincere thanks to you and each reviewer for the time, effort, and invaluable feedback provided. The constructive comments have given us a clearer perspective on areas for improvement. Most of the identified weaknesses can be addressed by conducting additional experiments and ablation studies, as well as by enhancing the related work section to clarify our key contributions. We hope that these revisions will effectively respond to your insights and further emphasize the robustness and novelty of our approach.

---

### Meta-Review · Area_Chair_AcSd · 2024-12-21

**Metareview:**

In this paper, the authors propose an LLM agent framework to simulate real users in recommendation systems. Reviewers found the idea interesting and easy to understand but raised concerns about the novelty of the approach and the insufficiency of experiments to support the claims. A key issue is whether the level of novelty is adequate, and reviewers generally believe it falls short of the standard required for publication in this conference.

**Additional Comments On Reviewer Discussion:**

The authors responded by clarifying the novelty, addressing the availability of the code, and committing to adding more experiments, including an ablation study. However, all reviewers have replied, expressing dissatisfaction with the responses.

---

### Decision · Program_Chairs · 2025-01-22

Reject